# MORSE: A Suite of Programmatically Controllable Multimodal Reasoning Environments with Steerable Difficulty

## Abstract

Despite rapid progress in vision language models, current multimodal reasoning *development pipelines* are limited by static training imagery, narrow task diversity, and benchmark saturation. We present **MORSE** ("Multimodal Reasoning Suite"), a programmatically controlled collection of *video* reasoning environments with *steerable difficulty* and *verifiable reasoning traces and answers*. The suite comprises: (i) **MORSE-∞**, a simulator that produces unlimited, difficulty steerable instances with reasoning traces; (ii) **MORSE-500**, a curated benchmark of 500 challenging videos covering six complementary reasoning categories and designed to retain headroom as models improve; and (iii) **MORSE-Agent**, which automates generation and curation to reduce human effort over time. Instances are produced via deterministic Python scripts (Manim, Matplotlib, MoviePy), generative video models, and curated real footage, exposing explicit controls over visual complexity, distractors, and temporal span. On **MORSE-500**, the strongest state of the art system achieves 23.6%, with a large gap to human performance at 55.4%, highlighting persistent deficits in abstract and planning categories. We release code, data, seeds, and the evaluation harness to support transparent, reproducible, and forward looking multimodal reasoning research.

## 1 Introduction

Progress in multimodal reasoning has been accelerated by ever-larger vision–language models (Bai et al., 2023; Hurst et al., 2024) and broader pretraining corpora (Li et al., 2025). Yet further advances are increasingly limited by the rigidity of available resources. Training corpora are static and rarely capture the temporal and interactive structure of the real world. Evaluation benchmarks quickly saturate, offering little room to probe new capabilities or expose failure modes (Lu et al., 2024). A sustainable path forward requires environments that can generate unlimited diverse problems, expose controllable difficulty, and provide verifiable reasoning traces that allow precise diagnostic analysis.

Existing resources fall short along several axes. First, most datasets focus on static images and short cues (Yue et al., 2023), underrepresenting temporal reasoning and multi-step dependencies. Second, practitioners lack access to scalable video-based resources with explicit difficulty control, making it hard to analyze model performance under varying levels of challenge. Third, current benchmarks generally do not provide ground-truth reasoning chains, leaving failure analysis and interpretability limited. Finally, evaluation sets quickly saturate, with little capacity to scale alongside improving models (Mirzadeh et al., 2024).

We introduce **MORSE** ("Multimodal Reasoning Suite"), a programmatically controllable collection of video reasoning environments with *steerable difficulty* and *verifiable reasoning traces and answers*. This suite contains three components:

- **MORSE-∞**, a data simulator that produces unlimited, difficulty steerable instances with reasoning traces and validated answers. This enables systematic analysis of model performance as task difficulty increases, and provides a rich source of verifiable reasoning chains that can be used for supervision or diagnostic evaluation.

- **MORSE-500**, a curated benchmark of 500 videos sampled from MORSE-∞, spanning six comple­mentary reasoning categories. It preserves headroom through controlled variation and seed based regeneration.
- **MORSE-Agent**, an agentic framework for automatically authoring new video generators. The agent proposes ideas, drafts Python code, receives structured feedback from a vision–language model critic, and iteratively refines its implementation until a functional generator emerges.

**Empirical findings and contributions.**    Evaluation of leading closed and open-source systems on MORSE-500 reveals substantial capability gaps, with the strongest models achieving only 23.6% accuracy compared to 55.4% human performance. Deficits are particularly pronounced in abstract reasoning (23.4% vs 37.5%) and planning tasks (5.0% vs 56.0%), indicating fundamental limitations in compositional reasoning and temporal integration.

This work establishes four foundational contributions to multimodal reasoning research:

- **Unified programmatic ecosystem**: The first integrated framework providing systematic control over complexity dimensions across training generation, rigorous evaluation, and autonomous cura­tion, with deterministic scripts enabling exact reproducibility and comprehensive trace supervision.
- **Scalable data generation**: MORSE-∞ delivers unlimited, difficulty-steerable instances with executable reasoning supervision, exposing explicit controls for visual complexity, distractor density, and temporal dynamics across six reasoning categories to support targeted curriculum design.
- **Challenging video reasoning benchmark**: MORSE-500 provides balanced assessment across temporal reasoning domains while preserving evaluation headroom through systematic difficulty scaling, with all generation scripts and seeds released for exact regeneration and extension.
- **Autonomous ecosystem evolution**: MORSE-Agent automates the complete development lifecycle from generation through difficulty calibration to adaptive curation, implementing a self-improving data ecosystem where enhanced model capabilities drive more sophisticated content generation.

## 2    MORSE ∞: INFINITE VIDEO GENERATION WITH DIFFICULTY CONTROL

MORSE-∞ is a generator of unlimited video reasoning problems with controlled difficulty and explicit step-by-step solutions. Unlike static datasets that quickly become outdated, MORSE- can continually produce new instances parameterized by complexity, temporal span, and distractor density. For each generated video, the underlying codebase also produces the ground-truth answer and a reasoning trace: a structured explanation of how the problem is solved, derived from the code itself.

### 2.1    PROGRAMMATIC GENERATION ARCHITECTURE

The generation pipeline synthesizes instances through deterministic Python scripts that define com­plete world models, render video sequences, and emit verified reasoning chains. Figure 2 illustrates the four-stage process: (1) parameter sampling from controlled difficulty distributions, (2) determinis­tic world state generation using fixed random seeds, (3) video rendering through Manim/Matplotlib with temporal coherence constraints, and (4) executable reasoning trace generation with comprehen­sive unit test validation. Each generator encodes domain-specific physics, maintains object persistence across frames, and ensures causal consistency between visual events and reasoning requirements. The modular architecture enables independent control over visual complexity (object count, occlusion patterns), temporal structure (sequence length, event timing), and reasoning depth (inference steps, compositional requirements) while maintaining semantic coherence.

### 2.2    REASONING TRACES FROM CODE

Since every task is generated from executable code, the complete solution is known by construc­tion. We implement templates that automatically translate code operations into step-by-step natural language explanations. For example, a pathfinding generator produces both the optimal path and a detailed reasoning trace. Figure 2 illustrates this process with a frozen lake navigation example (visualized in Figure 1 top row). The system generates the environment parameters, executes the

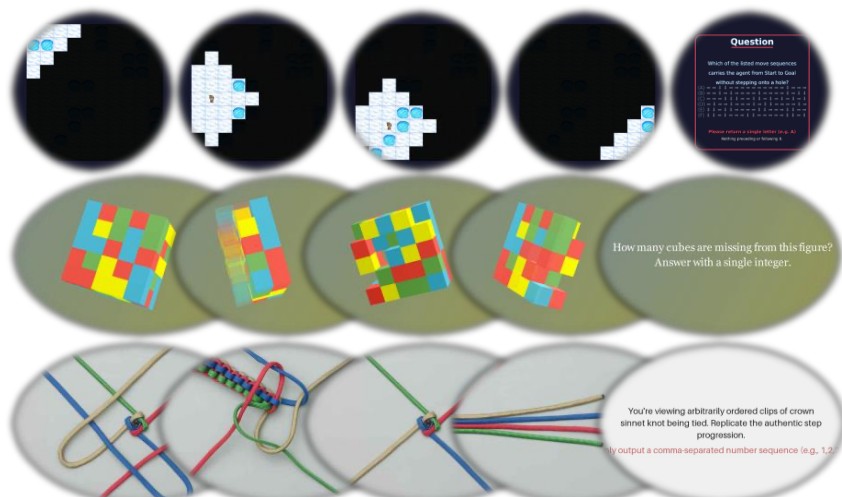

Figure 1: Examples from MORSE. Each row shows sampled frames (visualized as circles to reduce overlapping) from a different video task, ranging from maze navigation to rope tying sequence understanding.

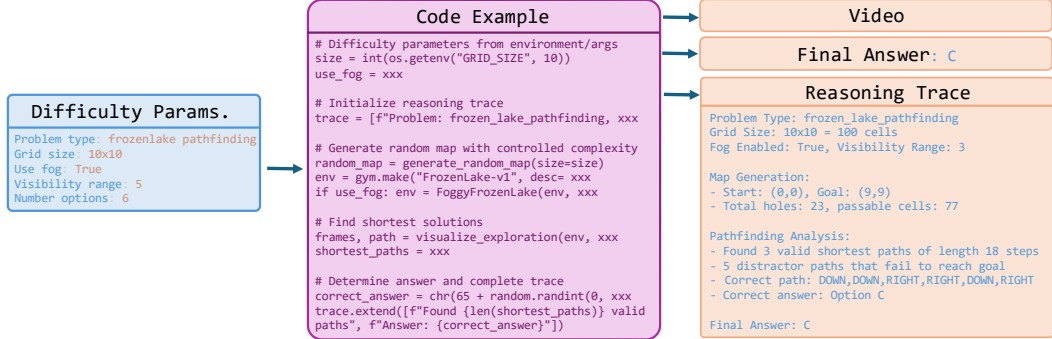

Figure 2: MORSE-∞ generation pipeline showing video synthesis with difficulty control, and reasoning trace generation with validated answer.

pathfinding algorithm, and simultaneously constructs a reasoning trace that explains each decision step.

The reasoning traces include intermediate states, transformation operations, and validation checks that verify consistency at each step. This approach eliminates human annotation errors and enables trace-level supervision during training, providing richer learning signals than traditional input-output pairs (Zelikman et al., 2022; Wang et al., 2022).

## 2.3 SYSTEMATIC DIFFICULTY CONTROL

Each reasoning category incorporates systematic difficulty variation through programmatically controlled parameters. Complexity is measured along five orthogonal dimensions: (1) entity complexity (2-15+ objects), (2) reasoning depth (1-5+ inference steps), (3) distractor density (minimal to 8+ irrelevant elements), (4) temporal complexity (static to dynamic concurrent processes), and (5) visual complexity (occlusion patterns, perspective changes, rendering variations).

Figure 4 demonstrates this systematic control across two task categories. As difficulty increases from left to right, performance consistently declines across all evaluated models, validating our parameterization. This controlled degradation enables us to sample appropriately challenging instances for MORSE-500.

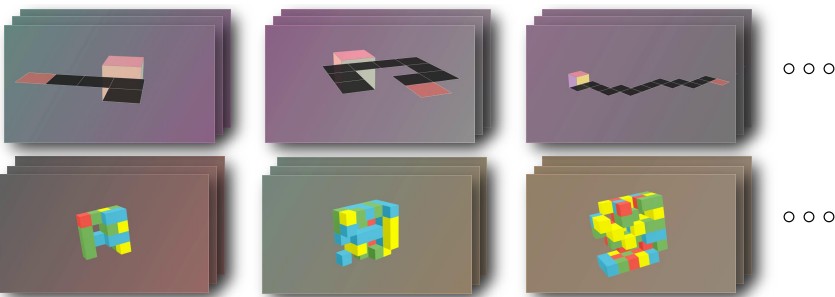

Figure 3: Programmatic difficulty control in MORSE. Each row demonstrates a series of videos with increasing complexity from left to right. For example the number of cube rotations and the size of the cubes controls the reasoning depth and visual complexity.

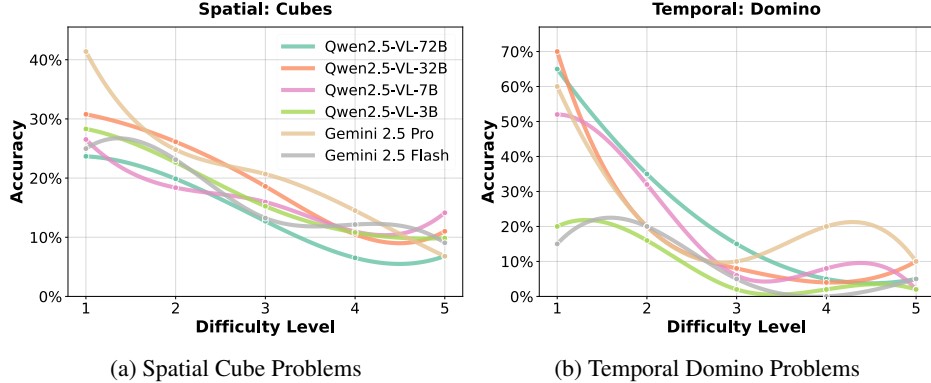

(a) Spatial Cube Problems          (b) Temporal Domino Problems

Figure 4: Model performance decreases systematically with increasing task difficulty across reasoning categories.

The programmatic foundation enables systematic expansion: new instances can be generated with specified difficulty profiles, and category-specific stress tests can be developed as model capabilities advance.

## 2.4 MULTI-DOMAIN REASONING COVERAGE

MORSE-∞ generates data across six complementary reasoning categories, each with specialized generators that maintain domain-specific constraints while respecting unified difficulty parameterization. **Mathematical reasoning** involves geometric transformations, algebraic relationships, and quantitative analysis with controlled complexity scaling. **Abstract reasoning** creates pattern recognition tasks based on ARC-AGI principles with systematic control over pattern complexity and transformation depth. **Spatial reasoning** synthesizes 3D transformation tasks, perspective changes, and spatial relationship problems with parameterized geometric complexity. **Temporal reasoning** produces sequence understanding tasks with controlled temporal dependencies and event complexity. **Physical reasoning** generates intuitive physics scenarios with systematic control over physical complexity and causal chain length. **Planning reasoning** creates multi-step goal-directed tasks with parameterized planning horizon and environmental complexity. Each domain generator maintains semantic coherence while ensuring systematic curriculum progression across all reasoning types, with domain-specific validation to ensure generated instances remain meaningful and solvable.

## 3 MORSE-500: EVALUATING MULTIMODAL REASONING LIMITS

Based on MORSE ∞, we sample 500 challenging problems to test the limits of current AI capable of multi-modal reasoning, and release this test set as a benchmark.

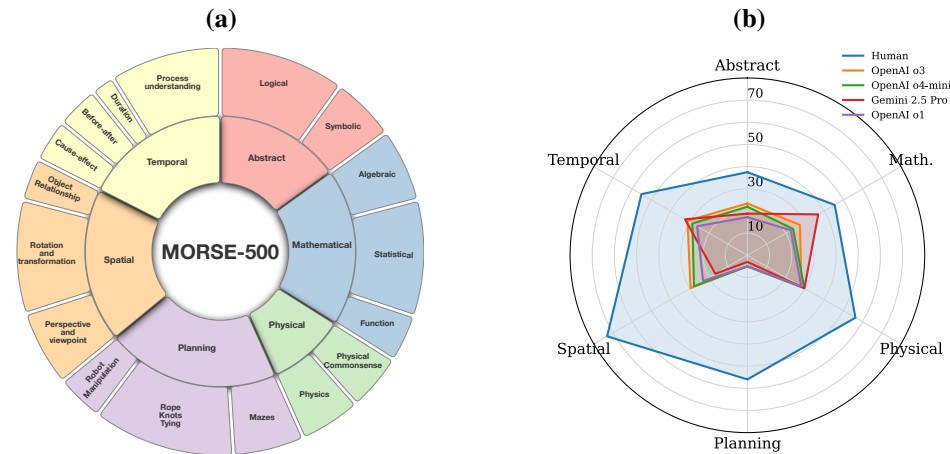

Figure 5: **(a)** Task distribution of MORSE. **(b)** Performance of the best-performing models on MORSE.

### 3.1 EVALUATION SETTINGS

**Models and Baselines.** We evaluate a diverse set of vision-language models spanning both proprietary and open-source architectures. Closed-source models include Gemini-2.5-Pro (Google, 2025), Gemini-2.5-Flash, Gemini-2.0-Flash variants, and Gemini-1.5-Pro from Google DeepMind, alongside OpenAI's o3 (OpenAI, 2025), GPT-4o (Hurst et al., 2024), o1 (Jaech et al., 2024), and o4-mini (OpenAI, 2025). Open-source models encompass the Qwen2.5-VL family (Bai et al., 2023) (3B, 7B, 32B, 72B) with and without quantization (AWQ), Qwen2.5-Omni-7B (Xu et al., 2025), LLaVA-NeXT-Video-7B (liu, 2024), MiniCPM-o-2_6 (Yao et al., 2024), InternVL3-8B (Zhu et al., 2025), and Gemma-3 (Gemma Team et al., 2025).

**Evaluation Protocol and Metrics.** Models with native video support received full video clips, while image-only models used frames sampled at 2fps with maximum 32-frame context. All models received the minimal prompt: `"Answer the question in this video."` with no few-shot examples to isolate intrinsic reasoning abilities. For models without video API support (OpenAI models), we provided downsampled image frames (512px maximum side length). We report accuracy as the primary metric, extracting answers using LLM-based parsing and string matching following MathVista (Lu et al., 2024).

### 3.2 QUANTITATIVE RESULTS

Table 1 presents accuracy across reasoning categories for all evaluated models. Overall performance remains substantially below human-level capabilities, with even the strongest systems averaging below 25% accuracy—a significant gap compared to human performance at 55.4%.

**Proprietary Model Performance.** Among proprietary models, OpenAI's o3 achieves the highest overall score of 23.6%, demonstrating relatively balanced performance across categories with particular strength in temporal reasoning (31.2%). Gemini-2.5-Pro follows closely at 21.8%, exhibiting notable proficiency in mathematical reasoning (36.9%) and temporal understanding (32.5%), but struggling significantly with abstract reasoning (18.8%) and planning tasks (3.0%). Interestingly, while Gemini-2.5-Flash achieves similar mathematical performance (35.7%), it shows markedly weaker performance in abstract reasoning (9.4%) and planning (1.0%), suggesting that model scale and optimization strategies significantly impact reasoning capabilities across different cognitive domains.

The performance patterns reveal interesting trade-offs: models optimized for mathematical reasoning tend to excel in structured, rule-based tasks but struggle with open-ended abstract reasoning. Conversely, models with stronger general reasoning capabilities (like o3) show more balanced performance but may sacrifice peak performance in specific domains.

Table 1: Accuracy (%) on MORSE across all six reasoning categories and overall average. Models are organized by closed-source and open-source categories.

| Model | All | Abstract | Math. | Physical | Planning | Spatial | Temporal |
|-------|-----|----------|-------|----------|----------|---------|----------|
| Human | 55.4 | 37.5 | 45.5 | 56.3 | 56.0 | 73.1 | 55.2 |
| **Closed-source Models** | | | | | | | |
| o3 | 23.6 | 23.4 | 27.4 | 28.1 | 5.0 | 29.6 | 31.2 |
| o4-mini | 22.2 | 21.9 | 23.8 | 29.7 | 5.0 | 27.8 | 28.7 |
| Gemini-2.5-Pro | 21.8 | 18.8 | 36.9 | 29.7 | 3.0 | 16.7 | 32.5 |
| o1 | 19.8 | 17.2 | 22.6 | 28.1 | 5.0 | 23.1 | 26.2 |
| Gemini-2.5-Flash | 19.2 | 9.4 | 35.7 | 28.1 | 1.0 | 24.1 | 18.8 |
| Gemini-1.5-Pro | 18.8 | 12.5 | 21.4 | 26.6 | 1.0 | 26.9 | 26.2 |
| GPT-4o | 17.4 | 17.2 | 20.2 | 34.4 | 4.0 | 12.0 | 25.0 |
| Gemini-2.0-Flash | 16.0 | 12.5 | 29.8 | 28.1 | 0.0 | 13.0 | 18.8 |
| Gemini-2.0-Flash-Lite | 14.2 | 17.2 | 21.4 | 21.9 | 2.0 | 14.8 | 12.5 |
| **Open-source Models** | | | | | | | |
| Qwen2.5-VL-72B | 17.8 | 6.2 | 21.4 | 34.4 | 1.0 | 22.2 | 25.0 |
| Qwen2.5-VL-32B-AWQ | 16.8 | 14.1 | 23.8 | 34.4 | 1.0 | 15.7 | 18.8 |
| Qwen2.5-VL-72B-AWQ | 16.4 | 12.5 | 11.9 | 29.7 | 2.0 | 27.8 | 16.2 |
| Qwen2.5-VL-32B | 15.6 | 9.4 | 19.0 | 29.7 | 2.0 | 16.7 | 21.2 |
| Gemma-3-27b | 14.6 | 20.3 | 20.2 | 25.0 | 1.0 | 13.0 | 15.0 |
| MiniCPM-o-2.6 | 11.6 | 4.7 | 10.7 | 23.4 | 1.0 | 16.7 | 15.0 |
| Qwen2.5-Omni-7B | 11.4 | 6.2 | 9.5 | 21.9 | 2.0 | 15.7 | 15.0 |
| Qwen2.5-VL-7B | 11.2 | 7.8 | 11.9 | 25.0 | 2.0 | 12.0 | 12.5 |
| InternVL3-8B | 7.8 | 6.2 | 6.0 | 14.1 | 1.0 | 11.1 | 10.0 |
| Qwen2.5-VL-3B | 7.6 | 9.4 | 3.6 | 18.8 | 1.0 | 9.3 | 7.5 |
| LLaVA-NeXT-Video-7B | 5.0 | 1.6 | 11.9 | 6.2 | 0.0 | 5.6 | 5.0 |
| Chat-UniVi-7B | 1.0 | 1.6 | 1.2 | 0.0 | 2.0 | 0.0 | 1.2 |

**Open-Source Model Analysis.** The open-source landscape demonstrates a clear scaling relationship between model size and reasoning performance. Among the Qwen2.5-VL family, the 72B model achieves 17.8% overall accuracy, substantially outperforming smaller variants (32B: 16.8%, 7B: 11.2%, 3B: 7.6%). However, quantization effects are mixed: while the 72B-AWQ model shows slightly lower overall performance (16.4%) compared to its full-precision counterpart, the 32B-AWQ variant actually outperforms the standard 32B model (16.8% vs. 15.6%), suggesting that quantization impacts vary with model scale.

Specialized models show domain-specific strengths: MiniCPM-o-2.6, despite lower overall performance (11.6%), demonstrates competitive physical reasoning capabilities (23.4%), indicating that targeted optimization can yield focused improvements. However, highly specialized models like LLaVA-NeXT-Video-7B, despite being designed for video understanding, achieve only 5.0% overall accuracy, highlighting the gap between video comprehension and video-based reasoning.

**Category-Specific Performance Patterns.** Performance varies dramatically across reasoning categories, revealing systematic weaknesses in current models. Mathematical reasoning shows the highest performance, with several systems exceeding 20% accuracy due to the structured nature of mathematical problems and their prevalence in training data. Physical reasoning demonstrates moderate competency (20-35% for top performers), suggesting intuitive physics concepts are partially captured in training paradigms, though the gap from human performance (56.3%) remains substantial. Spatial and temporal reasoning show moderate but inconsistent performance across models, with some displaying surprising deficits (e.g., Gemini-2.5-Pro's 16.7% spatial accuracy despite strong mathematical performance). Abstract reasoning proves most challenging, with even the best performers struggling to exceed 25% accuracy, suggesting fundamental limitations in pattern recognition, analogical thinking, and rule induction—core components of general intelligence. Most concerning, planning tasks show near-random performance across all models (0-5% accuracy), indicating critical gaps in multi-step reasoning and goal-directed behavior with significant implications for real-world deployment in autonomous systems.

**Implications and Model Limitations.** The uniformly low performance across all reasoning categories, particularly in abstract reasoning and planning, suggests that current multimodal models suffer from fundamental architectural limitations rather than mere training inefficiencies. The inability to perform multi-step reasoning, maintain temporal coherence, and engage in abstract pattern matching indicates that these models may be primarily engaging in sophisticated pattern matching rather than genuine reasoning.

Furthermore, the substantial human-model performance gaps (30+ percentage points in most categories) underscore that achieving human-level multimodal reasoning remains a significant challenge. The particularly poor performance on planning tasks raises questions about the suitability of current models for autonomous decision-making applications.

## 3.3 DIAGNOSTIC ANALYSIS OF MODEL LIMITATIONS

MORSE-500 enables systematic diagnosis of current model limitations across reasoning domains. We analyze performance on particularly challenging task categories to identify specific failure modes.

Figure 6 illustrates four demanding task types: Mazes require optimal pathfinding under partial visibility, testing temporal reasoning and spatial memory. Rope Knots involve sequence reconstruction from randomized tying steps with visual transformations. Physical Commonsense tasks distinguish realistic from AI-generated videos, probing physical intuition. ARC-AGI-2 adaptations test abstract pattern completion and counting under complex visual transformations.

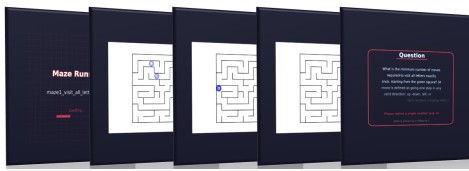

(a) Planning Reasoning - Maze with Different Endpoints

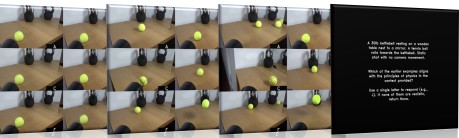

(b) Physical Commonsense - Tennis Ball Rolling

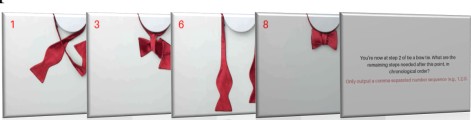

(c) Planning Reasoning - Knots of Tying a Bow Tie

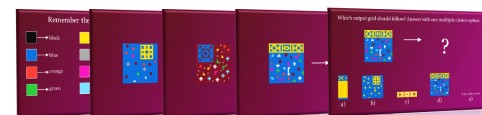

(d) Abstract Reasoning - ARC-AGI-2

Figure 6: Example videos for challenging tasks: Mazes, Rope Knots, Physical Commonsense, and ARC-AGI-2.

Table 2 reveals systematic limitations across all model categories. While humans achieve substantial performance (47.1-63.6% across tasks), even frontier models struggle dramatically: o3 reaches only 3.8-14.0%, and Gemini 2.5 Pro scores near-zero (0.0-4.2%). This performance gap highlights critical weaknesses in temporal reasoning, physical intuition, and abstract pattern recognition—fundamental capabilities for robust multimodal reasoning.

| Model | Mazes | Rope Knots | Physical Commonsense | ARC-AGI-2 |
|---|---|---|---|---|
| Human | 58.3 | 53.8 | 63.6 | 47.1 |
| o3 | 10.0 | 3.8 | 13.6 | 14.0 |
| Gemini 2.5 Pro | 0.0 | 1.3 | 4.2 | 0.0 |

Table 2: Performance of top frontier models across 4 challenging tasks.

These diagnostic results demonstrate MORSE-500's value in identifying specific architectural and training limitations, providing clear targets for future model development across reasoning categories.

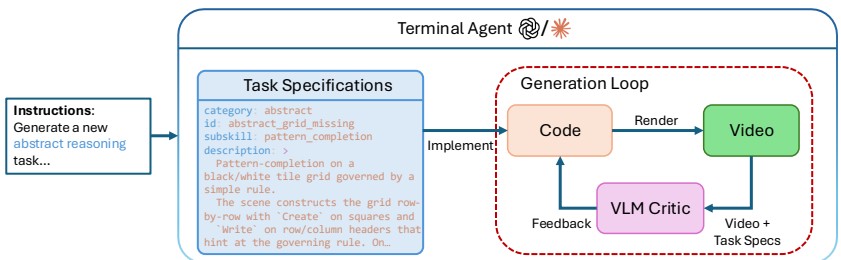

Figure 7: Workflow of MORSE-Agent. The agent proposes an idea, drafts code, receives feedback from a VLM critic, and refines its code iteratively until a stable generator is produced.

## 4 MORSE AGENT: AUTONOMOUS PROGRAM GENERATION

While MORSE-∞ provides an infinite stream of video reasoning tasks once generators are authored, building those generators is itself non-trivial. Each generator must define sampling rules, render coherent video scenes, and produce corresponding solution traces. To scale beyond hand-written scripts, we introduce MORSE-Agent, an agentic framework for automatically writing new video generators.

### 4.1 AGENTIC CODE GENERATION LOOP

MORSE-Agent is a terminal agent that operates in an iterative workflow inspired by program synthesis with feedback (Austin et al., 2021; Ellis et al., 2023):

1. **Idea Proposal** – The agent draws on the six-category taxonomy as scaffolding for proposing new generator ideas. The taxonomy is provided as input guidance, and example code serve as demonstrations.

2. **Initial Implementation** – Using example code snippets and templates, the agent drafts a Python program that implements the proposed generator.

3. **VLM Critic Feedback** – The draft is executed, and the resulting video along with the code are passed to a vision–language model critic. The critic evaluates correctness, coherence, and alignment with the intended reasoning category, and provides structured feedback. Here we use Gemini 2.5 Pro as the critic.

4. **Refinement** – The agent revises its code based on the critic's feedback, aiming to fix bugs, improve clarity, or better match the task description.

5. **Iteration** – Steps 2–4 repeat for several rounds until the generator consistently produces valid videos with correct answers and solution traces.

### 4.2 GENERATED TASK EXAMPLES

Figure 8 demonstrates MORSE-Agent's iterative refinement process on an abstract reasoning task, showing how the system identifies and corrects critical issues like premature answer revelation. MORSE-Agent reduces the manual burden of generator authoring and enables rapid expansion of the MORSE suite. By leveraging multimodal feedback, it creates generators that are both syntactically correct and semantically meaningful. However, its effectiveness depends on the quality of the VLM critic, and it may require several iterations to converge to a valid generator. MORSE-Agent should be viewed as an *enabler* of MORSE-∞, bootstrapping new generators efficiently.

## 5 RELATED WORK

**Controllable Complexity Evaluation.** Recent work reveals systematic limitations in reasoning models through controlled complexity manipulation. GSM-Symbolic (Mirzadeh et al., 2024) shows that LLMs exhibit significant variance when mathematical problem templates are systematically

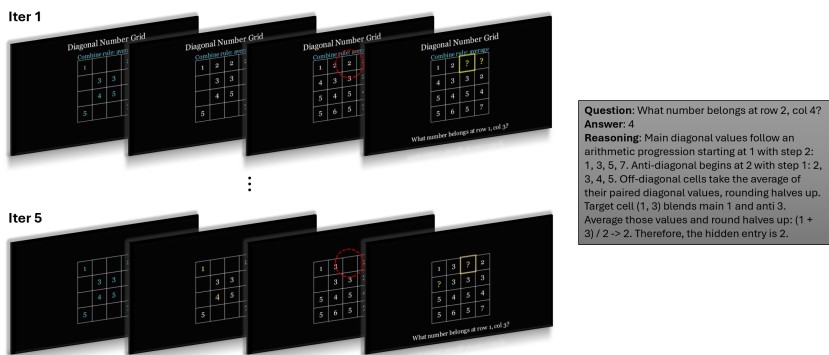

Figure 8: Example generation process for an abstract reasoning task. Code refinement is able to fix a critical issue with the answer being revealed in the middle of the video.

varied, exposing brittleness in seemingly robust capabilities. Shojaee et al.(Shojaee et al., 2025) use controllable puzzle environments to demonstrate "accuracy collapse" in Large Reasoning Models beyond specific complexity thresholds. OMEGA(Sun et al., 2025) evaluates three generalization axes: exploratory (complex instances within domains), compositional (combining distinct skills), and transformative (novel strategies). Game-RL (Tong et al., 2025) synthesizes verifiable game tasks with controllable difficulty across 30 games, demonstrating out-of-domain generalization from game-specific training. These approaches establish the foundation for our programmatic difficulty control in multimodal video reasoning.

**Multimodal Reasoning Benchmarks.** Existing benchmarks fall into two categories with complementary limitations. Static image benchmarks like MathVista (Lu et al., 2024), MMMU (Yue et al., 2023), and ARC-AGI (Chollet et al., 2025) provide strong reasoning challenges but ignore temporal dynamics and quickly saturate. Video understanding benchmarks like Video-MME (Fu et al., 2024), MVBench (Li et al., 2024), Mementos (Wang et al., 2024b), and (Song et al., 2025) incorporate temporal information but focus primarily on perception and retrieval rather than systematic reasoning evaluation. MORSE-500 bridges this gap by combining the systematic reasoning challenges of static benchmarks with the temporal complexity of video understanding, while providing programmatic difficulty control to prevent saturation. Our approach enables precise modulation of complexity dimensions and generates unlimited instances for continuous evaluation as models improve.

## 6 CONCLUSION

We introduce MORSE (Multimodal Reasoning Suite), a programmatically controlled collection of video reasoning environments with three integrated components: MORSE-$\infty$ for unlimited difficulty-steerable instance generation, MORSE-500 as a challenging benchmark with headroom for model improvement, and MORSE-Agent for automated generator creation. Our evaluation reveals substantial gaps in current AI capabilities. On MORSE-500, state-of-the-art systems achieve only 23.6% accuracy versus 55.4% human performance, with severe deficits in abstract reasoning (23.4% vs 37.5%) and planning tasks (5.0% vs 56.0%). These systematic failures indicate fundamental limitations in compositional reasoning and temporal integration. MORSE's programmatic foundation enables precise control over visual complexity, distractors, and temporal dynamics with verifiable ground truth. Unlike static benchmarks that saturate quickly, the suite's scalable difficulty generation ensures continued relevance as models advance, with modular design supporting extension to additional reasoning domains.

**Limitations.** Current generators emphasize synthetic environments; incorporating naturalistic scenarios while preserving verifiability remains challenging. MORSE-Agent depends on critic quality and may require iterative refinement. **Broader Impact.** MORSE enables targeted analysis of reasoning failures and supports development of more robust multimodal AI systems for real-world applications.

**LLM Usage**: Language models were used to improve text clarity and readability.

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
