CONTENTS

## A  MORSE-500 BENCHMARK OVERVIEW & MULTI-MODAL REASONING TAXONOMY

We construct MORSE-500 around six major categories of multi-modal reasoning, each designed with specific task instantiations and complexity controls, as detailed in Table A.1.

Our approach combines established benchmarks with novel adaptations to create challenging multi-modal scenarios that require both visual understanding and temporal reasoning.

**Task Design Methodology.** For each reasoning category, we employ a systematic approach to task creation that balances coverage and difficulty. We adapt existing datasets where appropriate (e.g., maze navigation (Ivanitskiy et al., 2023; Brockman et al., 2016), ARC-AGI (Chollet, 2019; Chollet et al., 2025)), introduce environmental challenges (e.g., fog effects, visual noise), and curate real-world scenarios (e.g., robotic manipulation sequences and rope knot tying procedures). Complexity is controlled through multiple dimensions specific to each category, including structural parameters (number of objects, plan length), environmental factors (visual noise, temporal irregularity), and cognitive demands (rule depth, interaction complexity).

**Novel Adaptations.** Several categories feature innovative task designs. For planning reasoning, we introduce fog effects to standard maze and FrozenLake environments (Brockman et al., 2016) and curate action sequence recognition tasks from robotic manipulation videos (Zhu et al., 2023; Wu et al., 2025; Haldar et al., 2023; Wu et al., 2023; Wang et al., 2023; Bahl et al., 2023). Physical reasoning incorporates a novel real-vs-generated video discrimination task that leverages the improving quality of video generation models. We use the videos in Physics IQ Benchmark Motamed et al. (2025) as real footage, we use image-to-video models, including Sora (OpenAI, 2024), Runnway Gen-3 (Germanidis, 2024), Kling 1.6 (Kling AI, 2025), Hailuo AI (Hailuo AI, 2025), Wan 2.1 (Wan Team et al., 2025), and Veo2 (Google DeepMind, 2024) to generate 5s videos conditioned on one video frame and textual description on the object and camera motion. Abstract reasoning repurposes ARC-AGI patterns into multiple-choice and free-form visual reasoning questions based on color patterns and spatial arrangements.

### A.1  DESIGN PRINCIPLES

The development of MORSE-500 was guided by four foundational principles aimed at addressing critical limitations in existing multimodal reasoning benchmarks while establishing a robust framework for systematic evaluation and future extensibility. Our design emphasizes **temporal-first evaluation** through video-based tasks that require genuine temporal understanding, **truly vision-centric assessment** where questions are embedded directly within visual content rather than provided as separate text, a **comprehensive reasoning taxonomy** grounded in established cognitive frameworks spanning six complementary reasoning categories, and **programmatic generation with scalable difficulty** enabling systematic complexity control and forward compatibility as model capabilities advance.

**Comprehensive Reasoning Taxonomy.**  MORSE-500 spans six complementary reasoning categories (see table A.1), each grounded in established cognitive science frameworks and designed to evaluate distinct cognitive capabilities essential for robust multimodal intelligence. Our taxonomy draws from the Cattell-Horn-Carroll (CHC) theory of cognitive abilities (Carroll, 1993), dual-process theory (Evans and Stanovich, 2013), and computational models of reasoning (Holyoak and Morrison, 2013). **Abstract reasoning** targets pattern recognition, logical inference, and symbolic reasoning associated with fluid intelligence, requiring operation at multiple levels of abstraction (Gentner, 1983). **Mathematical reasoning** evaluates fluid reasoning (Gf) and quantitative knowledge (Gq), assessing arithmetic operations, algebraic relations, and quantitative analysis through dynamic visualizations that integrate visual-spatial information with numerical processing (Dehaene, 2011). **Physical reasoning** examines intuitive physics understanding through object dynamics and causal interactions, bridging perceptual experience with abstract physical knowledge (McCloskey et al., 1983). **Planning reasoning** evaluates executive function and goal-directed behavior, emphasizing multi-step reasoning and sequential decision-making central to cognitive control theories (Miyake et al., 2000). **Spatial reasoning** corresponds to visual-spatial processing (Gv), testing 3D transformations, perspective understanding, and object relationships through tasks requiring mental model construction and manipulation (Shepard and Metzler, 1971). **Temporal reasoning** addresses sequence understanding

| Reasoning Category | Description |
|---|---|
| Abstract Reasoning (12.8%) | It targets *pattern recognition*, *logical inference*, and *symbolic reasoning* through adapted benchmarks and novel task designs. For logical reasoning, we repurpose ARC-AGI 2 patterns into multiple-choice questions and free-form responses based on color patterns, spatial arrangements, and cell counts, requiring rule induction from minimal examples and pattern extrapolation under abstract transformations. For symbolic reasoning, we design novel tasks including anagram word transformations, symbolic equation solving, and visual-textual symbol mapping challenges. Complexity is controlled by number of visual elements (4-25 cells), rule depth (1-4 nested transformations), color diversity (2-10 colors), symbolic abstraction level, and cross-modal symbol correspondence requirements. |
| Mathematical Reasoning (16.8%) | It evaluates *arithmetic operations*, *algebraic relations*, and *quantitative comparisons* through visual and textual integration. Tasks include dynamic word problems with chart interpretation, visual equation solving with geometric constraints, proportional comparisons across multiple data modalities, and geometric reasoning requiring spatial-numerical synthesis. Complexity is controlled through variable count (2-8 unknowns), operation depth (1-4 nested operations), visual noise levels, and cross-modal information density. |
| Physical Reasoning (12.8%) | It tests understanding of *object dynamics* and *causal interactions* governed by physical laws through simulation and real-world discrimination. Tasks involve predicting structural collapse in block towers, estimating force effects on object motion, forecasting collision trajectories, and a novel real-vs-generated video discrimination task where models must identify authentic physics from increasingly sophisticated generated alternatives. Complexity is determined by interaction complexity (1-8 interacting objects), diversity of physical principles (gravity, friction, momentum, elasticity), and realism of generated distractors. |
| Planning Reasoning (20.0%) | It emphasizes *multi-step*, *goal-directed reasoning* through adapted environments and real-world scenarios. Tasks include maze navigation with fog effects (adapted from standard maze datasets), FrozenLake traversal under partial observability, robotic manipulation sequence recognition from curated online videos, and rope-tying action ordering tasks. We test action sequencing, goal inference, and sequential decision-making with complexity defined by plan length (3-15 steps), environmental uncertainty (fog density, partial observability), constraint density, and branching factor (2-6 alternative paths). |
| Spatial Reasoning (21.6%) | It tests understanding of *object relationships*, *spatial transformations*, and *3D reasoning* across multiple viewpoints and reference frames. Tasks include mental rotation with occlusion handling, multi-view inference requiring perspective integration, spatial pathfinding through complex 3D environments, and relative positioning under dynamic transformations. Complexity is determined by number of objects (3-12), degree of transformation ($0°$-$360°$ rotations), scene dimensionality (2D/3D), and presence of visual distractors. |
| Temporal Reasoning (16.0%) | It assesses *sequence understanding* and *causal inference* over time through multi-frame visual narratives and process documentation. Tasks include event reordering from shuffled image sequences, duration comparison across parallel processes, future-state prediction in dynamic scenes, and cause-effect relationship identification in temporal chains. Difficulty is influenced by temporal irregularity (non-uniform time intervals), number of concurrent events (1-5), sequence length (4-20 frames), and presence of temporal red herrings. |

Table A.1: Definitions, task instantiations, and proportions of six multimodal reasoning categories in MORSE-500. Each category incorporates specific complexity controls and novel adaptations to create challenging multimodal scenarios.

and causal inference over time, evaluating temporal order tracking and future state prediction aligned with event segmentation theory (Zacks et al., 2007).

**Programmatic Generation with Scalable Difficulty and Forward Compatibility.** To ensure extensibility, reproducibility, and precise experimental control, all videos are generated through deterministic Python scripts utilizing established libraries including Manim for mathematical visualizations and 2D/3D object rendering and animation, Matplotlib for statistical graphics, MoviePy for image transforming effects, and video generative models for realistic scenario generation. This programmatic foundation enables fine-grained manipulation of complexity parameters including entity count, reasoning depth, distractor density, temporal dynamics (static to highly dynamic sequences), and visual complexity (minimal to high-noise environments).

A core innovation of MORSE-500 lies in its systematic difficulty progression that can evolve alongside model capabilities. Unlike static benchmarks that quickly saturate and become obsolete, MORSE-500

functions as a living evaluation framework where new instances can be generated with precisely specified complexity profiles. Difficulty scaling operates across multiple orthogonal dimensions: structural complexity (number of entities, interaction patterns), cognitive demands (reasoning depth, abstraction level), environmental challenges (visual noise, occlusion, temporal irregularity), and task-specific parameters (plan length for planning tasks, transformation complexity for spatial reasoning). The deterministic generation process ensures perfect reproducibility while supporting systematic difficulty scaling as model capabilities advance, enabling the identification of specific reasoning weaknesses and targeted evaluation of architectural improvements. This scalable architecture ensures that MORSE-500 remains diagnostically valuable as models improve, supporting the generation of arbitrarily challenging instances on demand while maintaining consistent evaluation standards and serving as an effective stress test for next-generation multimodal systems.

## A.2 DATASET STATISTICS

MORSE comprises 500 carefully curated video instances with embedded reasoning questions, systematically distributed across six complementary reasoning categories to ensure comprehensive cognitive coverage and balanced evaluation.

**Category Distribution and Strategic Allocation.** As shown in table A.1, the dataset employs a purposeful distribution across reasoning domains, with category allocation reflecting both cognitive importance and evaluation priorities: Spatial reasoning (21.6%, 108 instances) receives the largest allocation given its fundamental role in multimodal understanding; Planning reasoning (20.0%, 100 instances) emphasizes multi-step reasoning capabilities critical for autonomous systems; Mathematical reasoning (16.8%, 84 instances) covers structured problem-solving across arithmetic, algebraic, and geometric domains; Temporal reasoning (16.0%, 80 instances) evaluates sequence understanding and causal inference over time; while Abstract reasoning (12.8%, 64 instances) and Physical reasoning (12.8%, 64 instances) provide focused assessment of pattern recognition and physics-based inference respectively.

Within each category, tasks span multiple specialized subcategories: mathematical reasoning includes arithmetic operations, algebraic relations, geometric analysis, and quantitative comparisons; abstract reasoning encompasses pattern recognition, logical inference, and symbolic reasoning; spatial reasoning covers object relationships, spatial transformations, and 3D reasoning; temporal reasoning evaluates sequence understanding and causal inference over time; physical reasoning tests object dynamics, causal interactions, and physics laws; and planning reasoning examines multi-step reasoning and goal-directed problem solving.

**Video Characteristics.** Our dataset comprises 500 videos with a total duration of 3.1 hours and an aggregate file size of 1.4 GB. As shown inn Figure A.1, the videos exhibit considerable diversity in temporal characteristics, with durations ranging from 5.1 to 140.0 seconds (mean: 22.1s, median: 18.0s, std: 19.3s), indicating a predominance of short-form content with high information density. Frame rates vary significantly across the dataset, spanning from 5.0 to 60.0 FPS with a mean of 35.6 FPS and median of 30.0 FPS. Notably, 60 FPS emerges as the most frequent frame rate, reflecting modern high-quality video capture standards. In terms of spatial resolution, the dataset demonstrates a multi-modal distribution: 45.2% of videos are recorded in Full HD (1920×1080), while 27.0% are in standard definition (854×480), and 16.0% utilize an intermediate resolution of 800×533 pixels. The remaining videos span various resolutions including square formats (1920×1620, 1600×1600), collectively representing diverse recording devices and platform requirements. File sizes exhibit high variability, ranging from less than 0.1 MB to 68.2 MB (mean: 2.9 MB, median: 0.8 MB), with the substantial difference between mean and median suggesting a right-skewed distribution dominated by smaller files with occasional larger outliers. This heterogeneous composition reflects the natural diversity of developer-generated content across different software environments and conditions.

**Task Identification and Adaptation.** Beginning with our established six-category reasoning taxonomy, we systematically identify high-quality exemplars from existing benchmarks (Wang et al., 2024a; Chollet et al., 2025; Motamed et al., 2025) and expand them into dynamic video formats. For **abstract reasoning**, we adapt ARC-AGI pattern recognition tasks Chollet (2019) into animated sequences showing rule transformations over time, and included other tasks such as symbolic reasoning with anagram transformation. **Mathematical reasoning** draws inspiration from dynamic

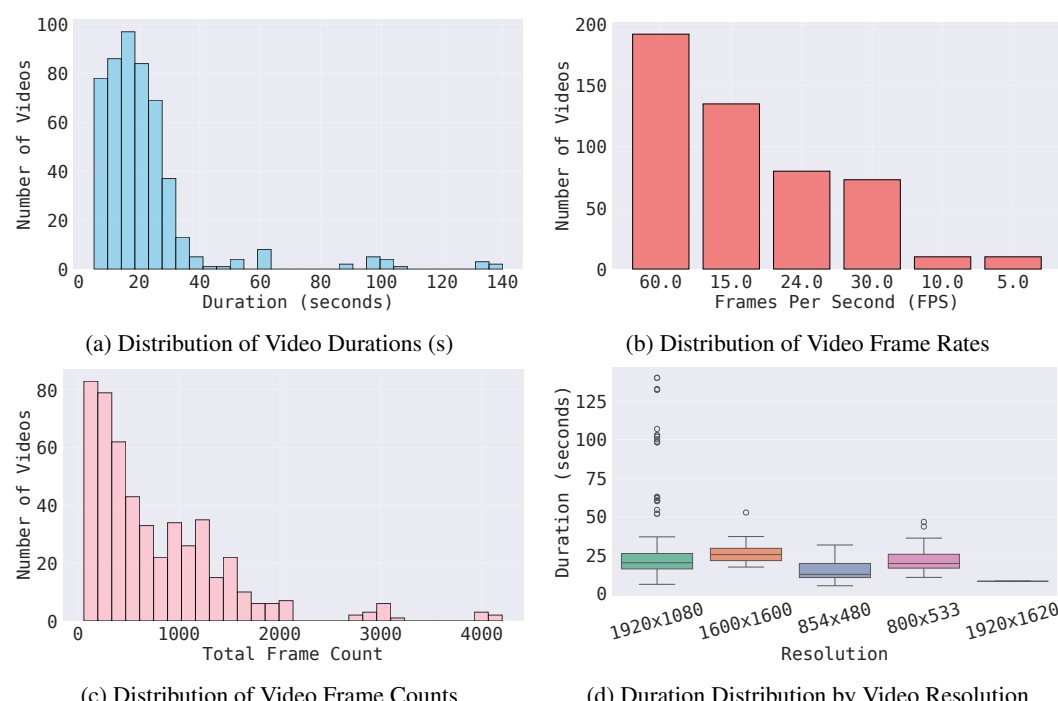

Figure A.1: Dataset statistics including duration, fps, frame count, and resolution.

visualizations in educational content, particularly the 3Blue1Brown YouTube channel's approach to mathematical explanation through animation, adapting static problems from math benchmarks (Lu et al., 2024) into temporal mathematical narratives. **Spatial reasoning** extends traditional mental rotation tasks from (Wang et al., 2024a) into continuous 3D transformations with occlusion and perspective changes. **Physical reasoning** incorporates real-world physics scenarios from established datasets (Motamed et al., 2025) while generating synthetic alternatives using state-of-the-art video generation models for discrimination tasks. **Planning reasoning** leverages maze and frozen-lake environment Brockman et al. (2016); Ivanitskiy et al. (2023), animated rope knots tying database (Grog, 2025), and robotic manipulation datasets (Zhu et al., 2023; Wu et al., 2025; Haldar et al., 2023; Wu et al., 2023; Wang et al., 2023; Bahl et al., 2023), transforming sequential action demonstrations into temporal reasoning challenges. **Temporal reasoning** creates novel sequence understanding tasks through procedural animations and cause-effect chains.

## A.3 DATASET COMPARISON

Recent advances in vision–language models (VLMs) have pushed the boundaries of perception and retrieval (Alayrac et al., 2022; Li et al., 2023; OpenAI, 2023), but robust reasoning remains elusive (Zhang et al., 2023; Lu et al., 2024). As these models are increasingly deployed in domains requiring inference, planning, and interaction—from embodied agents (Shridhar et al., 2023) to scientific assistants (Shen et al., 2023)—there is a growing need to evaluate and develop their capacity for genuine reasoning over multimodal inputs. This shift demands capabilities that go beyond recognition or retrieval, toward causal, temporal, abstract, and physically grounded understanding.

**Benchmark evolution—and persistent blind spots.** The trajectory of evaluation has mirrored model capabilities: early benchmarks focused on recognition (e.g., TextVQA (Singh et al., 2019), DocVQA (Mathew et al., 2021), OCR-VQA (Mishra et al., 2019)), then knowledge retrieval (e.g., MMMU (Yue et al., 2023), ScienceQA (Lu et al., 2022)), and more recently mathematical reasoning (Lu et al., 2024; Wang et al., 2024a; Zou et al., 2024), and physical reasoning (Chow et al., 2025; Qiu et al., 2025; Xiang et al., 2025). However, most of these datasets rely on static images and narrowly scoped question types, overlooking reasoning in dynamic, interactive environments where the ability to process sequences, anticipate outcomes, and generalize abstract patterns is essential.

| Benchmark | Math | Abstract | Spatial | Temporal | Physical | Planning |
|---|---|---|---|---|---|---|
| MathVista (Lu et al., 2024) | ✓ | | | | | |
| MathVision (Wang et al., 2024a) | ✓ | | | | | |
| DynaMath (Zou et al., 2024) | ✓ | | | | | |
| Mementos (Wang et al., 2024b) | | ✓ | ✓ | ✓ | | ✓ |
| HourVideo (Chandrasegaran et al., 2024) | | | ✓ | ✓ | | |
| LongVideoBench (Wu et al., 2024) | | | | ✓ | | |
| EMMA (Hao et al., 2025) | ✓ | ✓ | ✓ | | ✓ | |
| PhysBench (Chow et al., 2025) | | | ✓ | ✓ | ✓ | |
| PHYBench (Qiu et al., 2025) | ✓ | | | | ✓ | |
| SeePhys (Xiang et al., 2025) | ✓ | | | | ✓ | |
| **MORSE-500 (Ours)** | ✓ | ✓ | ✓ | ✓ | ✓ | ✓ |

Table A.2: Comparison of different benchmarks.

**Limitations of current benchmarks.**    Despite recent progress, today's reasoning benchmarks suffer from three structural limitations:

- **Static modality bias:** Most benchmarks rely on single-frame images, ignoring the temporal evolution and causality inherent to many real-world tasks.
- **Narrow reasoning spectrum:** They often focus heavily on math word problems (Lu et al., 2024; Wang et al., 2024a; Zou et al., 2024), underrepresenting reasoning types such as spatial logic, temporal inference, physical causality, abstraction, and multi-step planning.
- **Rapid saturation:** Many benchmarks are quickly saturated by current models (Lu et al., 2024), offering little diagnostic signal once performance plateaus.

Moreover, current benchmarks often conflate reasoning with perception and retrieval (Zhang et al., 2023), making it difficult to assess whether models are genuinely reasoning or merely pattern-matching. This highlights the need for a principled evaluation framework that systematically varies difficulty while explicitly controlling for perceptual and knowledge-based confounds. Our benchmark addresses the above limitations. See A.2 for a comparison between different benchmarks.

## B    MORSE-500 MORE QUALITATIVE RESULTS

We present reasoning CoTs from various frontier VLMs: Qwen3-235B-A22B (queried via their official website), o3, Gemini 2.5 Pro. Samples are selected to represent each type of reasoning category. For failure modes, incorrect reasoning is highlighted in red.

### B.1    ABSTRACT REASONING

Here we show the full thinking trace for Qwen3-235B-A22B and Gemini-2.5-Pro-Preview 05-06. For o3 we only present some of the key CoTs due to its excessively long reasoning and function calling. The full trace can be found in https://chatgpt.com/share/68301c0a-78a4-8003-aeff-eeb5cee85221.

---

**Abstract Reasoning - ARC-AGI2**

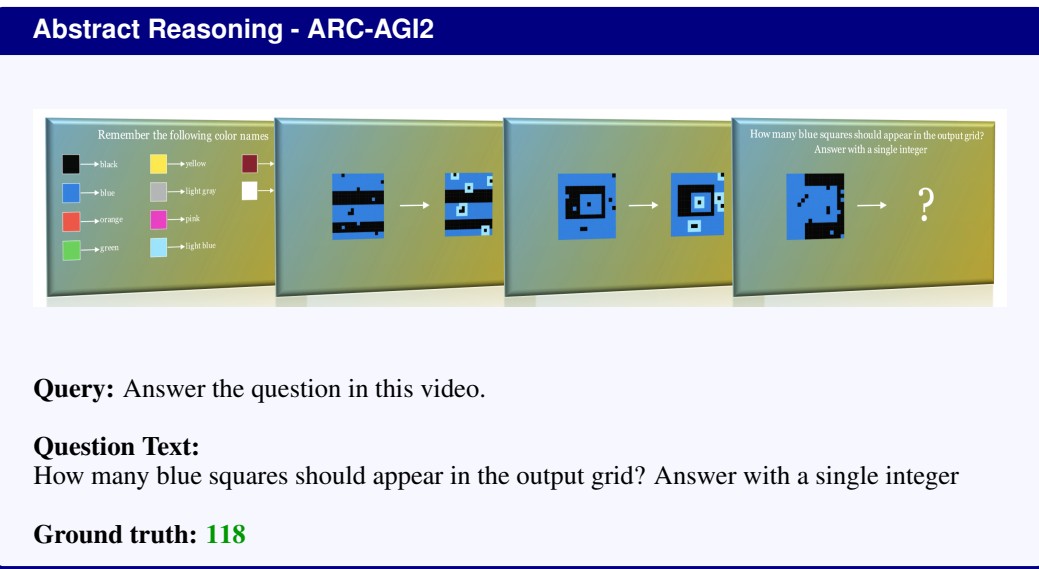

**Query:** Answer the question in this video.

**Question Text:**
How many blue squares should appear in the output grid? Answer with a single integer

**Ground truth: 118**

---

**Qwen3-235B-A22B**

*Thinking*

Okay, let's see. The video starts by showing a color chart with different colors labeled. Then there are some pixelated images that change from one to another. The last frame asks how many blue squares should appear in the output.

First, I need to figure out the pattern or rule that's being applied to the pixelated images. Let me look at the examples given. The first input is a blue square with some black pixels around it. The output shows the same blue square but with some blue pixels added in the corners. Then another example where the input is a blue square with a smaller blue square inside, and the output adds more blue pixels around the smaller one. The third example has a blue shape that's more spread out, and the output seems to fill in some areas.

Wait, maybe the rule is that each time, the number of blue squares increases by one. Let me check. In the first example, the input has one blue square, and the output has two. The second input has two blue squares, and the output has three. The third input has three blue squares, so the output should have four. But wait, the last frame shows an input with three blue squares and asks how many should be in the output. So following this pattern, the answer would be four.

But I should make sure there's not another pattern. Maybe it's about the arrangement or the direction of the squares. But looking at the examples, each time the number increases by one. So the answer is probably four blue squares.

The video presents a series of pixelated images where the number of blue squares increases by one in each subsequent output. Observing the examples:

1. First Input : 1 blue square → Output : 2 blue squares.

2. Second Input : 2 blue squares → Output : 3 blue squares.

3. Third Input : 3 blue squares → Output : ?

Following the established pattern, the next output should contain 4 blue squares .

**Answer:** 4 (Incorrect)

---

## o3

*Thought for 9m 17s*

I'm thinking about how to inspect the frames in the video. The goal here is to understand what the video shows, so maybe I'll examine the first few frames or perhaps the last frame where the question appears. It seems likely that the text was graphically created, so I'll check for that. I'll load the video and start with the frames to see what stands out and where the relevant visual or text elements are located.

[. . . ] (1615 words omitted)

It seems like the task involves counting items, possibly within a grid of shapes, and the question could be something like, "How many black squares are there?" The answer might be a number, like "4." To be sure, I'd probably need some clarification, but the user didn't ask for that. So, I think it would be best to use OCR to detect text in the frames. I'll process the frames at a low resolution to make it easier and redefine the video path as needed.

[. . . ] (2921 words omitted)

It seems the user might be expecting a direct answer as part of an evaluation. Based on analysis, I think "3" is a reasonable guess, but I'm cautious because I can't visually confirm the content in the video. Even though I could extract frames, the user simply asked for an answer, not video frames. I'll mention
*Done*

---

3 (Incorrect)

## Gemini-2.5-Pro-Preview

**Interpreting Visual Data**

I'm working on understanding the video's core question. I've noted the color names and the grid examples, hoping to extract a pattern or rule. Now, I'm focusing on the input/output grid pairs, trying to find a correlation between them. Perhaps they encode color mapping rules? I'm examining each grid, seeking transformations.

**Unveiling Grid Transformations**

I'm now zeroing in on the hypothesized rule involving isolated black pixels on a blue background. I'm actively testing the proposed transformation rules with the example grids. I'm carefully analyzing how the light blue borders are drawn around the isolated black pixels, while paying close attention to the preservation of existing black regions. I'm checking if the proposed rule accurately explains the observed transformations in the examples.

**Revising Rule Assumptions**

[. . . ] (1719 words omitted)

**Counting Blue to Light Blue**

I've determined there are five trigger pixels, each causing 8 blue pixels to become light blue. My previous misinterpretation of "isolated" black pixels led to some confusion. With confidence, I'm ready to calculate the total number of light blue squares.

**Decoding Pixel Patterns**

I'm now counting the total number of light blue squares generated. The test input has five isolated black pixels; each triggers an 8-pixel change to light blue based on Example 2 logic. I am now certain of the solution.

- - - - - - - - - - - - - - - - - - - - - - - - - - - - - - - - - - - - - - - - - - - - - - - - - - - - - - - - - - -

The rule for the transformation, derived from the examples, is as follows:

1. Identify Trigger Pixels: A black pixel in the input grid is a "trigger pixel" if and only if all 8 of its surrounding neighbors (horizontally, vertically, and diagonally) are blue. Black pixels that are part of larger black structures (i.e., have at least one black neighbor) are not triggers and remain unchanged (they are "inert").

2. Transformation Rule:
   - Each trigger black pixel remains black in the output.
   - All 8 of its neighboring blue pixels (which, by definition of a trigger pixel, must all be blue) change to light blue in the output.
   - Blue pixels that are not neighbors of any trigger black pixel remain blue.
   - Inert black pixels remain black and do not cause any changes to their neighbors.

Let's apply this to the test input grid. The grid is 10x10. 'K' denotes black, 'B' denotes blue.

Input Grid:
Row 0: K K K K K K K K K K
Row 1: K K K K K K B B B B
Row 2: K K K K K K B K B B
Row 3: K K K K K K B B B B
Row 4: B B B B B B B B B B
Row 5: B K B B K B B K B B
Row 6: B B B B B B B B B B
Row 7: K K K K K K B B B B
Row 8: K K K K K K B K B B
Row 9: K K K K K K B B B B

Step 1: Count initial blue squares.

Row 1: 4 blue
Row 2: 3 blue
Row 3: 4 blue
Row 4: 10 blue
Row 5: 7 blue
Row 6: 10 blue
Row 7: 4 blue
Row 8: 3 blue
Row 9: 4 blue
Total initial blue squares = 4 + 3 + 4 + 10 + 7 + 10 + 4 + 3 + 4 = 49.

Step 2: Identify trigger black pixels.
We need to find black pixels (K) whose 8 neighbors are all blue (B).

- K at (2,7): Neighbors: (1,6)B, (1,7)B, (1,8)B, (2,6)B, (2,8)B, (3,6)B, (3,7)B, (3,8)B. All 8 are blue. This is a trigger pixel.

- K at (5,1): Neighbors: (4,0)B, (4,1)B, (4,2)B, (5,0)B, (5,2)B, (6,0)B, (6,1)B, (6,2)B. All 8 are blue. This is a trigger pixel.

- K at (5,4): Neighbors: (4,3)B, (4,4)B, (4,5)B, (5,3)B, (5,5)B, (6,3)B, (6,4)B, (6,5)B. All 8 are blue. This is a trigger pixel.

- K at (5,7): Neighbors: (4,6)B, (4,7)B, (4,8)B, (5,6)B, (5,8)B, (6,6)B, (6,7)B, (6,8)B. All 8 are blue. This is a trigger pixel.

- K at (8,7): Neighbors: (7,6)B, (7,7)B, (7,8)B, (8,6)B, (8,8)B, (9,6)B, (9,7)B, (9,8)B. All 8 are blue. This is a trigger pixel.

Other black pixels (e.g., at (0,0), (1,0)-(1,5), etc.) are part of larger black structures and have black neighbors, so they are not trigger pixels. There are 5 trigger black pixels.

Step 3: Determine how many blue squares change to light blue.
Each trigger black pixel causes its 8 blue neighbors to change to light blue. We need to check if the 3x3 regions of influence (centered on each trigger pixel) overlap.

- Region 1 (around (2,7)): Rows 1-3, Cols 6-8.

- Region 2 (around (5,1)): Rows 4-6, Cols 0-2.

- Region 3 (around (5,4)): Rows 4-6, Cols 3-5.

- Region 4 (around (5,7)): Rows 4-6, Cols 6-8.

- Region 5 (around (8,7)): Rows 7-9, Cols 6-8.

These regions are all disjoint. Therefore, the sets of blue pixels changing to light blue are also disjoint. Number of blue squares changing to light blue = 5 trigger pixels * 8 blue neighbors/trigger = 40.

Step 4: Calculate the number of blue squares remaining in the output.
Number of blue squares in output = (Initial blue squares) - (Blue squares changed to light blue)
Number of blue squares in output = 49 - 40 = 9. So, 9 blue squares should appear in the output.

Final Answer: The final answer is 9 (Incorrect)

Figure B.1: Example failure case in Abstract Reasoning using ARC-AGI pattern recognition. Three frontier models attempt to count blue squares in a transformed grid following pattern rules. Despite detailed reasoning traces, all models fail: Qwen incorrectly applies a simple increment pattern (answer: 4), o3 struggles with holistic visual understanding and provides most verbose yet ineffective reasoning leading to an arbitrary guess (answer: 3), and Gemini performs complex rule analysis but miscounts the final result (answer: 9). The correct answer is 118, demonstrating the challenge of visual pattern recognition and rule application in abstract reasoning tasks.

## B.2 MATHEMATICAL REASONING

Table B.2 illustrates representative model outputs on challenging examples from MORSE, and more examples can be found in Appendix B. We observe consistent failure patterns in tasks requiring abstract reasoning and multi-step planning, where even large-scale models falter at integrating temporal cues or executing compositional logic.

Smaller models often rely on superficial visual features, anchoring on salient but irrelevant distractors. In contrast, more capable models—such as Gemini 2.5 Pro and o3—attempt partial reasoning chains but frequently fall short of arriving at the correct solution. These qualitative trends reinforce the quantitative gaps reported earlier and highlight the need for better memory, inference, and grounding in dynamic visual contexts.

---

**Mathematical Reasoning - 3D Extrema Plot**

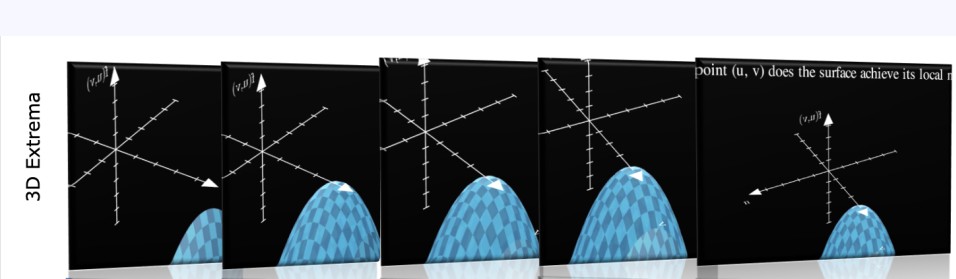

Figure B.2: Example of mathematical reasoning failure across state-of-the-art models. All three models hallucinate the maximum at the origin, whereas the ground truth is $(0, 4)$.

**Query:** Answer the question in this video.

**Question text:** At what point $(u, v)$ does the surface achieve its local maximum?

**Ground truth:** **(0,4)**

---

**Response from Qwen3-235B-A22B**

To determine the point $(u, v)$ at which the surface achieves its local maximum, let's analyze the given 3D surface plot step by step.

The surface is a 3D plot with coordinates $(u, v, z)$, where $z$ represents the height of the surface. The goal is to find the point $(u, v)$ where the surface reaches its local maximum.

The surface appears to have a peak or a highest point. The peak is visually identifiable as the point where the surface rises the most and then begins to descend in all directions.

From the plot: the surface has a distinct peak. The peak is located near the origin of the $u$–$v$ plane, but it is not exactly at the origin. The surface slopes downward in all directions from this peak.

The peak appears to be at a point where both $u$ and $v$ are close to zero but not exactly zero. Based on the symmetry and the shape of the surface, the peak is likely at a point where $u$ and $v$ are small positive or negative values.

The exact coordinates of the peak cannot be determined precisely without numerical data, but visually, the peak is very close to the origin. Therefore, the point $(u, v)$ where the surface achieves its local maximum is approximately:
**(0, 0)**

**Response from o3 (summarized)**

The model begins by sampling video frames at regular intervals (e.g., 1 fps or 0.5 fps) using OpenCV:

```
import cv2, numpy as np, matplotlib.pyplot as plt
video_path = '/mnt/data/LocalExtreme3DQuestion1.mp4'
cap = cv2.VideoCapture(video_path)
fps = cap.get(cv2.CAP_PROP_FPS)
total = int(cap.get(cv2.CAP_PROP_FRAME_COUNT))
duration = total / fps
sample_rate = 30  # 2 fps
frames = []
for i in range(0, total, sample_rate):
    cap.set(cv2.CAP_PROP_POS_FRAMES, i)
    ret, frame = cap.read()
    if ret:
        frames.append((i, frame))
cap.release()
print(len(frames))  # 15 frames
```

It then inspects the sampled frames to locate the overlay text asking, "At what point $(u, v)$ does the surface achieve its local maximum?", and attempts to zoom in for clarity. Finally, it visually estimates the peak coordinates.
**Answer:** **(0,0)**

**Response from Gemini-2.5-Pro-Preview (summarized)**

The video displays a 3D surface, which appears to be an elliptic paraboloid opening downwards. The question asks for the point $(u, v)$ where the surface achieves its local maximum.

The axes are labeled:
• One horizontal axis is labeled 'u'.
• The other horizontal axis is labeled 'v'.
• The vertical axis is labeled 'f(u,v)' (implied by the function notation at the top, though only (v,u)f is visible).
A local maximum is a point on the surface that is higher than all its immediate neighbors. For a smooth, downward-opening paraboloid like the one shown, the local maximum is unique and occurs at its vertex.

By visually inspecting the graph, the vertex (the highest point) of the paraboloid is located directly above the origin of the $u$–$v$ plane. The origin is where the $u$-axis and $v$-axis intersect, which corresponds to $u = 0$ and $v = 0$.
Therefore, the surface achieves its local maximum at the point $(0, 0)$.

## Mathematical Reasoning - Radius Increase

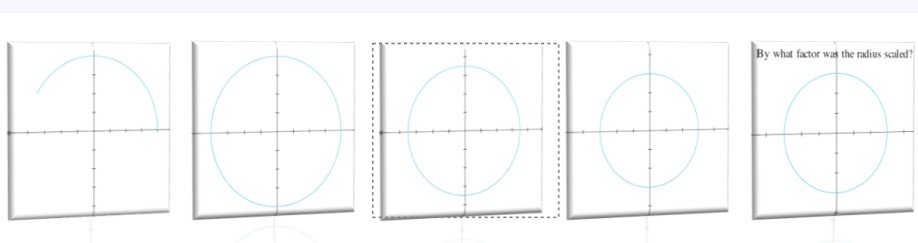

**Query:** Answer the question in this video.

**Question Text:** By what factor was the radius scaled?

**Ground truth: 0.75**

### Qwen2-235B-A22B

Okay, let's see. [...]

First, examine the starting figure: a horizontal baseline marked with tick marks, and a semicircle drawn above this line. The baseline represents the diameter of that semicircle.

Next, the semicircle is transformed into a full circle. Crucially, the baseline's length does not change, so it now equals the diameter of the **entire** circle.

A semicircle is exactly half of a full circle. To turn it into a full circle, the vertical dimension (i.e. the radius) must double while the horizontal diameter stays fixed. Hence the radius is scaled by a factor of 2.

**Answer: 2**

### o3

The user-supplied video (8 s at 60 fps) appears to show a circle that grows rapidly in the first second and then stays large. To uncover the question, I

- **Sampled frames** at 1 fps and ran `pytesseract` OCR. From frame 360 onward the overlay text consistently reads: "By what factor was the radius scaled?"
- **Detected circles** in each frame with a Hough transform. Early frames (#20–30) contain several concentric circles; the largest reliable radius is about 331 px. From frame 40 onward the dominant radius jumps to about 961 px.
- **Computed the ratio**: $\frac{961}{331} \approx 2.90$. Small measurement noise and pixel quantisation suggest the intended factor is exactly 3.

Hence the video shows the original circle instantly expanding so its radius triples.

**Answer: 3**

### Gemini-2.5-Pro-Preview

Okay, let's see. [...]

**00:00 – 00:01**

An upper semicircle is drawn. It meets the $x$-axis at roughly $x = -4$ and $x = +4$, with its apex at $(0, 4)$. That implies a full circle of radius 4 centred at the origin.

**00:01 – 00:03**
The animation completes the lower semicircle, forming a full circle that still spans $x = -4$ to $x = +4$ and $y = -4$ to $y = +4$. Thus the final circle's radius is also 4.

**Key observation:**
Both the "original" (implied by the first semicircle) and the final circle share the same radius, so no size change occurred.

**Scaling calculation:**

$$\text{Scaling factor} \;=\; \frac{\text{Final radius}}{\text{Original radius}} = \frac{4}{4} = 1.$$

**Answer:** 1

Figure B.3: Example failure case in Mathematical Reasoning involving radius scaling. Three frontier models attempt to determine the scaling factor when a semicircle transforms into a full circle. All models fail to correctly identify the geometric transformation: Qwen incorrectly reasons about doubling the radius (factor of 2), o3 uses image analysis to detect circle expansion (factor of 3), and Gemini claims no size change occurred (factor of 1). The correct answer is 0.75, representing the radius reduction needed when completing a semicircle into a full circle with the same diameter.

### B.3 PHYSICAL REASONING

Here we show the full thinking trace for Qwen3-235B-A22B and Gemini-2.5-Pro-Preview 05-06. For o3 we only present some of the key CoTs due to its excessively long reasoning and function calling. The full trace can be found in `https://chatgpt.com/share/683022c2-1400-8003-8f66-94f26ddaa311`. All models struggle with distinguishing the physical nuances of different scenarios.

---

**Physical Reasoning - Tennis Rolls Towards Mirror**

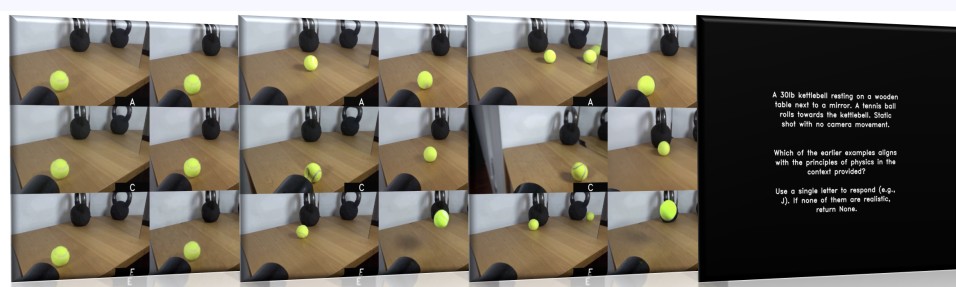

**Query:** Answer the question in this video.

**Question Text:**
A 30lb kettlebell resting on a wooden table next to a mirror. A tennis ball rolls towards the kettlebell. Static shot with no camera movement.

Which of the earlier examples aligns with the principles of physics in the context provided?

Use a single letter to respond (e.g., J). If none of them are realistic, return None.

**Ground truth: E**

---

**Qwen3-235B-A22B**

*Thinking*
Okay, let's see. The question is asking which of the earlier examples aligns with the principles of physics in the context provided. The setup is a 30lb kettlebell resting on a wooden table next to a mirror, and a tennis ball rolls towards the kettlebell. The shot is static with no camera movement.

First, I need to look at the images labeled A to F and figure out which one makes sense physically. Let me go through each image step by step.

In image A, there's one tennis ball moving towards the kettlebell. That seems straightforward. If a ball rolls towards a stationary object, it should hit it unless something else happens.

Image B shows the ball bouncing off the kettlebell. But wait, if the kettlebell is heavy (30lb) and stationary, the ball would just bounce off with some force. However, since the table is wooden and the surface might be smooth, the ball could roll away after hitting the kettlebell. But in B, the ball is shown mid-air, which might not be accurate because the collision would cause it to bounce, but maybe not that high. Also, the position of the ball relative to the kettlebell needs to make sense.

Image C has two balls. One is near the kettlebell, and another is rolling towards it. But the original setup mentions only one tennis ball. So maybe C is incorrect because there are two

---

balls here. Unless the first ball split or something, but that's not physical. So C might be out.

Image D shows two balls again, one near the kettlebell and another further away. Similar to C, this introduces an extra ball, which isn't part of the original scenario. So D is probably not correct.

Image E has one ball near the kettlebell and another smaller one rolling towards it. Again, two balls, so same issue as C and D. Unless the small one is a reflection, but the mirror is on the right side, and the reflection would show the ball's image, not another ball. So E is likely wrong.

Image F shows one ball rolling towards the kettlebell, similar to A, but the position is slightly different. The ball is closer to the kettlebell here. But does the path make sense? If the ball is rolling straight towards the kettlebell, it should hit it. However, in F, the ball is still moving towards it, so maybe it's just a matter of timing. But the question is about which image aligns with physics.

Wait, the key point here is that the kettlebell is heavy and stationary. When the tennis ball collides with it, the ball should bounce back because the kettlebell doesn't move much. So in image B, the ball is shown bouncing off, which would be expected. However, the height of the bounce might be an issue. If the collision is elastic, the ball would bounce back with the same speed, but the height depends on the angle and surface. But since the table is flat, the ball might not bounce very high. Alternatively, if the ball hits the kettlebell and then rolls away, that's also possible.

But looking at image B, the ball is in the air, which suggests it bounced. However, if the collision is head-on, the ball would reverse direction but stay on the table. So maybe image B is not accurate because the ball is shown too high in the air. Alternatively, if the ball hit the edge of the kettlebell and bounced sideways, that could be possible. But the images don't show the direction of the bounce.

Alternatively, image F shows the ball still approaching the kettlebell, which would be before the collision. So if the question is about the moment after the collision, then B would be the answer. But the problem is that the question says "a tennis ball rolls towards the kettlebell. Static shot with no camera movement." So the image should capture the state after the ball has rolled towards the kettlebell. If the ball has already collided, then B is the result. But if the ball is still rolling towards it, then A or F.
Wait, the initial setup is that the ball is rolling towards the kettlebell. The static shot is taken when the ball is in motion. So the correct image would be one where the ball is in the process of rolling towards the kettlebell, not after the collision. But the options include images where the ball has already hit the kettlebell (like B) or is still approaching (A, F).

But the question is which example aligns with physics. If the ball is rolling towards the kettlebell, then the physical principle is that the ball will continue moving until it hits the kettlebell. So the image should show the ball in motion towards the kettlebell. However, in the images, A and F show the ball moving towards the kettlebell, while B shows it bouncing off.

But the problem is that the question is asking which of the earlier examples (A-F) aligns with the principles of physics. The correct answer would be the one where the ball is moving towards the kettlebell, as that's the described scenario. However, if the ball has already hit the kettlebell, then B would be the result, but the setup says the ball is rolling towards it, implying that the collision hasn't happened yet.

Wait, the description says "a tennis ball rolls towards the kettlebell. Static shot with no camera movement." So the image should capture the ball in the act of rolling towards the kettlebell. Therefore, the correct image would be one where the ball is in motion towards the kettlebell, not after the collision. So images A, C, E, F show the ball moving towards the

kettlebell, while B and D show after the collision.

But among A, C, E, F, which ones are physically correct? The original setup has one ball. So C, D, E have two balls, which is incorrect. So only A and F are left.

In A, the ball is moving towards the kettlebell. In F, the ball is closer but still moving towards it. Both are possible, but the question is which one aligns with physics. Since the ball is rolling, the image should show the ball in a position that's consistent with its motion. However, without knowing the exact timing, it's hard to say. But the answer is likely A or F.

Wait, but the problem might be that in some images, the ball's position relative to the kettlebell is impossible. For example, if the ball is too close to the kettlebell in F, but given that it's rolling, it's possible.

Alternatively, the answer could be B if the collision is considered. But the setup says the ball is rolling towards the kettlebell, not that it has collided. So the static shot is of the ball in motion towards the kettlebell. Therefore, the correct image would be A or F.

But looking back at the images, in A, the ball is on the left side, moving towards the kettlebell. In F, the ball is closer to the kettlebell but still moving towards it. Since the question is about which example aligns with physics, both A and F are possible, but perhaps the answer is F because it's closer, indicating the ball is in the process of rolling. However, without more information, it's hard to tell.

Wait, the text after the images says: "A 30lb kettlebell resting on a wooden table next to a mirror. A tennis ball rolls towards the kettlebell. Static shot with no camera movement." So the correct image should show the ball in motion towards the kettlebell. The images that show the ball moving towards the kettlebell are A, C, E, F. But C, E have two balls, which is incorrect. So A and F.

Between A and F, which one is correct? The answer might be F because the ball is closer to the kettlebell, indicating it's further along in its roll. But the problem is that the question is about which image aligns with physics. Since the ball is rolling, the image should show the ball in a position that's consistent with its motion. However, all images are static, so they just capture a moment.

Alternatively, the answer could be B if the collision is part of the scenario. But the setup says the ball is rolling towards the kettlebell, not that it has collided. So the correct image is one where the ball is in motion towards the kettlebell. Therefore, the answer is A or F.

But looking at the images again, in A, the ball is on the left, moving towards the kettlebell. In F, the ball is closer but still moving towards it. The problem is that the answer is supposed to be a single letter. The user might expect that the correct answer is B because after the collision, the ball bounces, but the setup says the ball is rolling towards the kettlebell, not after the collision.

Wait, the question is phrased as "Which of the earlier examples aligns with the principles of physics in the context provided?" The context is a 30lb kettlebell on a table next to a mirror, a tennis ball rolls towards the kettlebell, static shot with no camera movement. So the image should show the ball in the process of rolling towards the kettlebell. Therefore, the correct image is A or F.

But why are there multiple images? Maybe the answer is F because the ball is closer, indicating it's further along in its path. However, without knowing the exact timing, it's hard to say. Alternatively, the answer could be None if none of the images are correct. But that seems unlikely.

Wait, looking at image B, the ball is shown in the air, which would be the result of the collision. But if the ball is rolling towards the kettlebell, the collision hasn't happened yet. So B is af-

ter the event. The correct image should be before the collision. Therefore, the answer is A or F.

But the problem is that the user might expect the answer to be B because that's the result of the physics (the ball bouncing off). But the setup says the ball is rolling towards the kettlebell, not that it has collided. So the answer should be an image where the ball is in motion towards the kettlebell.

Given that, the answer is either A or F. But which one? The images are labeled A to F, and the user might have intended one of them. Since the ball is rolling, the image should show the ball in a position that's consistent with its motion. However, all images are static, so they just capture a moment.

Alternatively, the answer could be None if none of the images correctly depict the scenario. But that's not likely.

Wait, the text after the images says: "Use a single letter to respond (e.g., J). If none of them are realistic, return None." So if none of the images are correct, the answer is None.

But let's re-examine the images.

Image A: One ball moving towards the kettlebell. This is possible.

Image B: Ball bouncing off. This would be after the collision.

Image C: Two balls. Incorrect.

Image D: Two balls. Incorrect.

Image E: Two balls. Incorrect.

Image F: One ball moving towards the kettlebell, closer than in A.

So between A and F, which is correct? The answer depends on the exact moment captured. Since the question is about which aligns with physics, both A and F are possible, but the answer is likely F because it's closer, indicating the ball is in the process of rolling. However, without more information, it's hard to be certain.

Alternatively, the answer could be B if the collision is considered part of the scenario. But the setup says the ball is rolling towards the kettlebell, not that it has collided.

Therefore, the correct answer is either A or F. But since the user expects a single letter, and considering that F shows the ball closer to the kettlebell, which is more in line with the action of rolling towards it, the answer is F.

Wait, but in the images, F shows the ball still moving towards the kettlebell, just closer. So yes, F is a valid depiction. Therefore, the answer is F.

- - - - - - - - - - - - - - - - - - - - - - - - - - - - - - - - - - - - - - - - - - - - -

The correct answer is **F**.

**Explanation:**
The scenario describes a tennis ball rolling towards a stationary 30lb kettlebell on a wooden table. The static shot must capture the ball in motion towards the kettlebell, not after a collision. Images C , D , and E are invalid due to the presence of two balls, violating the setup. Images A and F show one ball moving towards the kettlebell. While both are plausible, F depicts the ball closer to the kettlebell, aligning better with the dynamic of rolling motion. Image B incorrectly shows the ball bouncing off the kettlebell, which would occur after the collision, contradicting the described action. Thus, F most accurately reflects the physical

scenario.

**Answer:** F (Incorrect)

---

### o3

*Thought for 7m 42s*

I need to set up some code to sample frames from a video and possibly display them for analysis. To do this, I'll use Python with libraries like cv2 or moviepy. I'll start by extracting a few frames and gathering metadata, just for analysis. No need to dive into advanced stuff just yet; this is more about laying the groundwork. I'll focus on being clear on the process and use it to guide what comes next!
*Analyzed*

```
import cv2, os, json, math, numpy as np, base64, PIL.Image \
as Image, io, textwrap
```

I need to write some code to sample frames from a video, maybe display them for analysis. I'll use Python with libraries like cv2 or moviepy. First step: I'll need to load the video, then extract metadata and a few frames. This will help in analyzing how the data changes over time. It's just the initial step, so no fancy stuff just yet! Let's keep things simple and focused for now.

```
video_path = \
"/mnt/data/physical_reasoning_physical_commonsense_physics_iq_0156.mp4"
cap = cv2.VideoCapture(video_path)
if not cap.isOpened():
    print("Could not open video")
else:
    fps = cap.get(cv2.CAP_PROP_FPS)
    frame_count = int(cap.get(cv2.CAP_PROP_FRAME_COUNT))
    duration = frame_count / fps if fps else 0
    print("fps", fps, "frame_count", frame_count, "duration", duration)

    # output: fps 30.0 frame_count 240 duration 8.0
```

To start, I'll write code to sample frames from a video, maybe display them for analysis. I'll use Python, with libraries like cv2 or moviepy, to load the video and extract frames at specific times (0s, 2s, 4s, 6s). I could save frames to a file or possibly convert them to ASCII, but displaying text using PIL might be easier. Another option: try OCR, though it's a bit heavy for now. The alternative might be analyzing the audio for speech.

[. . . ] (2040 words omitted)

After showing the scenarios, the text on the final screen has a question and a list of scenario descriptions. The OCR only captures the first scenario, with others cut off. We need to capture the entire screen at 6.5 seconds, save the image, and extract more text. I'll crop the grayscale image and apply text recognition to capture everything. The question seems to ask which earlier example aligns with physics principles, so I should focus on descriptions that refer back to them.

[. . . ] (6297 words omitted)

I'm stuck. I was thinking of binarizing the image to locate the "J" template, but it seems like my search for the letter didn't work. At this point, the easiest solution might be to say there's no match, but the task still needs a letter. Maybe the letter corresponds not to the final scene but to earlier examples? The scenario description at the end might just summarize

prior content. It looks like the task involves matching principles of physics, not necessarily focusing on a specific letter.

[...] (516 words omitted)

I'll calculate the pixel difference between frame 0 and 3500 for each tile to detect movement. The middle rows seem to have the highest motion, possibly due to a ball scenario. Tile (1,3) stands out because of its high motion, but it might be an unrealistic scenario. I'll inspect it closely and save the tile image for further analysis. Let's see what we get from this!

*Analyzed*

```
r,c=1,3
y=r*tile_h
x=c*tile_w
tile=frame[y:y+tile_h,x:x+tile_w]
cv2.imwrite("/mnt/data/tile_13.png", tile)

# output: True
```

Let's calculate the pixel

*Done*

- - - - - - - - - - - - - - - - - - - - - - - - - - - - - - - - - - - - - - - - - - - - - - - - - -

**Answer:** A (Incorrect)

---

### Gemini-2.5-Pro-Preview

**Assessing Video Relevance**

I'm focusing on the core physics principles at play: the tennis ball's momentum transfer to the kettlebell, the influence of gravity, and the mirror's reflective properties. My current line of thought is to methodically eliminate videos that blatantly misrepresent these factors. This process will highlight those videos with accurate depictions of the physics, and I will be able to rule out each video based on whether it adheres to real-world physics principles.

**Refining the Evaluation Process**

I'm now refining the evaluation process. I'll need to carefully examine each video, starting with the ball's initial motion. The static table and mirror setup offer clear physical reference points. I will be looking closely at the ball's behavior after impact to determine if there's any movement, and if the reflection in the mirror appears accurate with respect to all of the criteria. My primary focus is to identify any deviations from these basic physics principles, particularly in the impact dynamics and reflective properties. Pinpointing Plausible Examples

My current focus is on the impact dynamics and mirror reflections. I'm carefully analyzing Example C, which appears most promising so far. The ball rebounds upon impact, and the kettlebell stays put, which is good. The mirror reflection aligns with the real-world physics, but I'll continue evaluating for anything that appears unrealistic. I'm focusing on Examples D, E, and F next, for any other acceptable answers.

**Determining Plausible Outcomes**

I'm now zeroing in on Example C. Its adherence to the basic physics principles is promising. The rebound of the tennis ball and the kettlebell's immobility are both realistic. The mirror reflection appears consistent, but I will still carefully assess for any other deviations. I'll also evaluate if the bounce of the ball is correct, and not an exaggeration. The focus is to look for any subtle discrepancies, even though it may seem accurate at first glance.

> **Identifying the Correct Video**
>
> I'm now fully immersed in the examples, having systematically assessed each based on the principles of momentum and reflection. Example C stands out as the most accurate representation. My focus now is solidifying this assessment. I'm double-checking the video's details: the initial roll, the impact dynamics (rebound and kettlebell immobility), and, crucially, the mirror reflection. I will confidently conclude that Example C adheres most closely to the real-world physics principles at play.
>
> - - - - - - - - - - - - - - - - - - - - - - - - - - - - - - - - - - - - - - - - - - - - - - -
>
> **Answer:** C (Incorrect)

Figure B.4: Example failure case in Physical Reasoning requiring discrimination between realistic and unrealistic physics scenarios. Models must identify which video shows plausible physics for a tennis ball rolling toward a 30lb kettlebell. Despite lengthy reasoning traces, all models fail: Qwen focuses on ball-kettlebell interaction timing but selects the wrong option (F), o3 struggles with video analysis and provides minimal justification (A), and Gemini attempts systematic physics analysis but misidentifies the realistic scenario (C). The correct answer is E, highlighting difficulties in understanding physical plausibility in dynamic scenes.

## B.4 Planning Reasoning

### Planning Reasoning - Robot Manipulation

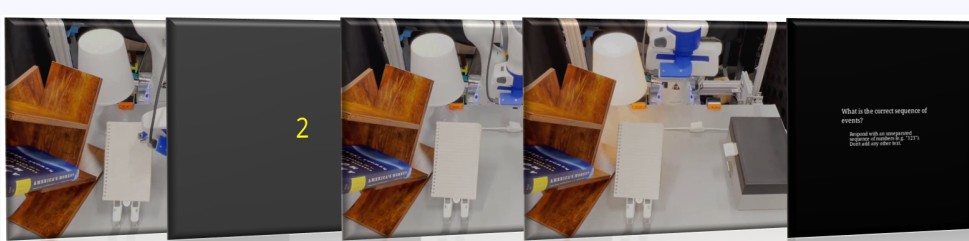

**Query:** Answer the question in this video.

**Question Text:** What is the correct sequence of events?

Respond with a comma-separated sequence of numbers (e.g. "1,2,3..."). Don't add any other text.

**Ground Truth:** 3,2,1

### Qwen2-235B-A22B

Okay, let's see. [...] (44 words omitted)

First, I'll look at each frame. Frame 1 shows the robot arm reaching towards a notepad that's held by two white clips. The lamp is turned off. In Frame 2, the robot arm is now holding the notepad, and the lamp is still off. Then Frame 3 shows the robot arm moving away, and the lamp is now on.

So the sequence seems to be: the robot picks up the notepad (Frame 1 to Frame 2), then turns on the lamp (Frame 2 to Frame 3). But wait, the lamp being on in Frame 3 might be a separate action. Let me check again.

In Frame 1, the lamp is off. In Frame 2, the lamp is still off while the robot is picking up the notepad. Then in Frame 3, the lamp is on. So the correct order would be picking up the notepad first (Frames 1-2), then turning on the lamp (Frame 3). Therefore, the sequence should be 1, 2, 3.

But wait, maybe the lamp turning on is part of the same action? No, because in Frame 2 the lamp is still off. So the robot first picks up the notepad (1-2), then turns on the lamp (3). So the correct sequence is 1, 2, 3.

**Answer:** 1,2,3 (Incorrect)

### o3

It looks like I need to extract frames from the last few seconds of the video to identify the question. I'll try using a library to load the video file and extract those frames. Once I have the frames, I might then try OCR (if available) or simply read the text manually. I can't show images to the user, but I can inspect the output visuals privately to find the question. Let's proceed with this method and check if OCR works.

[...] (586 words omitted)

It looks like the question is focused on the order of events, and earlier in the video, they may have labeled events with numbers or shown clips. The answer could be a sequence like "321." To find that, I should look at frames earlier in the video where labels such as "1)", "2)", or "3)" appear. I'll focus on frames between 300 and 400 to try and capture those numbered events, doing OCR on every 30th frame to reduce noise.

```
cap = cv2.VideoCapture(video_path)
numbers_found = []
for fi in range(total_frames):
    if fi %
        continue
    cap.set(cv2.CAP_PROP_POS_FRAMES, fi)
    ret, frame = cap.read()
    if not ret:
        continue
    gray = cv2.cvtColor(frame, cv2.COLOR_BGR2GRAY)
    text = pytesseract.image_to_string(gray, config='--psm 6')
    text_clean = text.strip()
    if any(digit in text_clean for digit in ['1','2','3']):
        numbers_found.append((fi, text_clean))
cap.release()
numbers_found[:20]
```

I'm looking for number overlays in frames, focusing on detecting digits like '1', '2', and '3'. If I sample every second of frames, I've found '1' in the first frame, '2' at frame 120, and '3' in frame 270. These could mark the segments, but the real question is about the original sequence of events, not just the labels. It seems the user might want to reorder the segments to their original order, 1, 2, 3.

The frames are labeled, starting with '1' at frame 0, which suggests no shuffle in the original order. However, considering the earlier script, the frame numbers correspond to shuffled segments, meaning that '1' could appear earlier in the sequence due to the shuffle. The labels reflect this shuffled order, not the original one, so the task is to identify the correct chronological order based on the original segmentation. The user probably expects an answer based on the original event order, though it's tricky without the key directly from the shuffle script.

[...] (237 words omitted)

The labels '1', '2', and '3' appear in order at frames 0, 120, and 270, respectively, suggesting the sequence is "123". There's no earlier '3' or '2' found, and sequence alignment seems confirmed. I've reviewed frames carefully, with '1' appearing first, then '2', and '3' last. Therefore, I'm confident that the answer is "123".

**Answer:** 1,2,3 (Incorrect)

---

**Gemini-2.5-Pro-Preview**

The video displays three segments, labeled 1, 2, and 3. We need to determine the correct chronological sequence of the events shown in these segments.

Segment 1 (0:00-0:03): Shows the robot arm already in contact with a notebook, drawing a line from its right to its left (right to left on the page from the perspective of something being written conventionally), and then lifting its "pen." This is an action of writing/drawing.
Segment 2 (0:04-0:08): Shows the robot arm performing an action nearly identical to Segment 1. It's in contact with the notebook, draws a line similarly, and lifts its "pen." This is also an action of writing/drawing.
Segment 3 (0:09-0:12): Shows the robot arm initially to the right of the notebook. It then moves over the notebook and positions its "pen" down towards the paper, as if to begin

writing. This is a preparatory action, an approach to start writing.

Chronological order of events:
The robot arm must first approach the notebook and position itself to write. This action is shown in Segment 3.
After positioning, the robot arm performs the writing action. This is shown in Segment 1.
If there is a subsequent writing action, or if Segment 2 represents a continuation or a second stroke, it would follow. Segment 2 shows a writing action very similar, if not identical, to Segment 1.

The preparation/approach (Segment 3) must come first. Then, the writing occurs. Segment 1 shows a writing action. Segment 2 also shows a writing action. If this is a second stroke or a continuation, it would follow Segment 1 (assuming the numerical labels 1 and 2 imply an order for these similar actions). Thus, the logical sequence of these clips to form a coherent event is: Segment 3 (approach) -¿ Segment 1 (write first part) -¿ Segment 2 (write second part/repeat). The sequence of numbers is 312.

Answer: 3,1,2 (Incorrect)

Figure B.5: Example failure case in Planning Reasoning involving temporal sequence understanding. Models are provided a shuffled video from the MimicPlay dataset Wang et al. (2023), and tasked with determining the correct chronological order of the robot's actions (switching off light and shelving notebook) from shuffled video segments labeled 1, 2, 3. While Qwen correctly identifies individual frame states, all models fail at sequential reasoning: o3 and Qwen both provide the displayed order (1,2,3) without recognizing the shuffling, while Gemini attempts logical sequencing but incorrectly orders the preparation and execution phases (3,1,2). The correct sequence is 3,2,1.

B.5   SPATIAL REASONING

## Spatial Reasoning - Count the Cubes

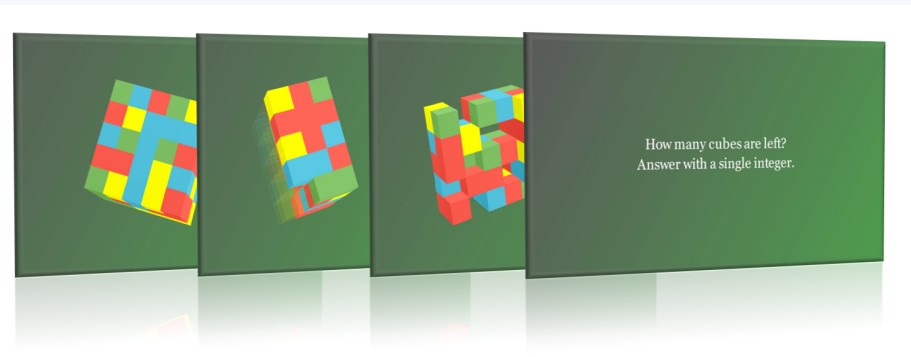

How many cubes are left?
Answer with a single integer.

**Query:** Observe the following structure. How many cubes are left?
Answer with a single integer.

**Ground Truth:** 42

### o3

The user uploaded a video named "count_2.mp4" and wants an answer to a question from it. However, this video is not accessible through regular tools here. I'll need to use a Python-based approach to extract frames or audio, and maybe process them with available tools like OpenCV and ffmpeg. My goal is to examine the content of the video to locate the question and provide the answer.

[. . . ] (92 words omitted)

The user likely uploaded a video "count_2.mp4" with a counting question. We don't need to show the frames unless necessary, but I will inspect them to understand the question. I'll open the video and check a few frames to see what it's asking. I'll use tools available, like imageio or OpenCV, to load the video and analyze it. Once we have an idea from the frames, we can provide a clearer answer to the user's question.

```
import cv2
cap = cv2.VideoCapture('/mnt/data/count_2.mp4')
frames = []
fps = cap.get(cv2.CAP_PROP_FPS)
frame_count = int(cap.get(cv2.CAP_PROP_FRAME_COUNT))
duration = frame_count / fps if fps else None
duration
```
[. . . ] (400 words omitted)

Okay, let's give this another shot. I wonder if the question in the video is actually contained in the audio. We can extract the audio using ffmpeg (if available) and then perhaps transcribe it. Though I don't have access to speech recognition libraries offline, I might try to manually guess what's being said. The file is likely small, so it shouldn't take too long to inspect it. Time to load the audio and see what we can figure out!

[. . . ] (1695 words omitted)

Okay, I'm going to plot the red pixel counts over time and print the list of values out. This should help me figure out any patterns, like when red objects appear or disappear. It's a simple way to visually track the changes across frames. Hopefully, that will give me a clearer idea about the objects in the video and help me make a more accurate guess. Let's see what the data shows!

```
for t, rp in zip(times, [red_pixels(f) for f in frames_sampled]):
    print(t, rp)
```

I'm going to plot the red pixel counts over time and print the list. The red pixels appear around 4s to 11s, which could indicate a red object across several frames. To answer the question, I'll focus on counting the red objects. Let's take a look at frame 8 (around t=8), which has a high count of red pixels. I'll check for connected components of the red clusters there to help identify distinct red objects. This should give us a better idea.

[...] (1701 words omitted)

It seems like there's a pattern of red clusters in the frames where the red objects appear. I also notice green clusters, though they might represent a background or axis. Maybe the question relates to how many times the red ball hits the green one, but I'm not totally sure. The pattern of red clusters isn't strictly increasing or decreasing, it seems a bit erratic. I'll print the counts to analyze the sequence. The animation could be showing some kind of calculation too. Let's see what we find.

*Done*

**Answer**: 6

---

### Qwen3-235B-A22B

Okay, let's see. The question is asking how many cubes are left after some structure is observed. The video shows a 3D cube made up of smaller cubes. The main cube is divided into smaller sections, each colored differently.

First, I need to figure out the total number of small cubes in the structure. The cube appears to be a 3x3x3 Rubik's cube, which typically has 27 small cubes. But wait, maybe it's not a standard Rubik's cube. Let me check again.

Looking at the images, the cube is shown from different angles. Each face has a mix of colors, but the structure seems to have layers. If it's a 3x3x3 cube, there are 3 layers along each axis. So, 3*3*3 equals 27. But sometimes, these puzzles might have fewer cubes if some are missing. However, in the images, the cube looks complete, with all sides filled.

Wait, but the question says "how many cubes are left." Maybe some cubes are removed? But in the video frames provided, the cube is intact. There's no indication of any cubes being taken away. The last frame just asks the question. So perhaps the answer is the total number of small cubes in the structure as shown.

If it's a 3x3x3 cube, the answer would be 27. But let me make sure. Sometimes, these questions can be tricky. For example, if the cube is hollow, the number would be less. But the images show a solid cube with colored faces. Each small cube is visible on the surface, and since it's a 3D structure, the inner cubes are also present.

Another way to count: each layer (front, middle, back) has 9 cubes (3x3). Three layers would be 27. Yes, that makes sense. So unless there's a trick here, like overlapping or something, the answer should be 27. But the video might be showing a different structure. Wait, looking at the third image, the cube is rotating, and it seems like there are some gaps or

maybe overlapping cubes? No, actually, it's just the perspective making it look like that. The cube is still a solid 3x3x3.

Therefore, the number of cubes left is 27. The answer should be 27.
**Answer**: 27

---

**Gemini-2.5-Pro-Preview**

Assessing Cube Quantity

I'm currently focused on determining the missing cubes within the 3x3x3 structure. The video's rotation makes visualization a bit tricky, but I'm mentally parsing the layers to count the vacancies. I plan to refine the count by checking for overlapping voids and hidden blocks.

Refining Cube Count

I'm now breaking down the structure further, layer by layer, considering each face. At 0:04, it looks like the top layer is missing one cube in the center, and the rotation is helping me clarify the positions. The video timing is making it hard to follow. Analyzing each face individually could provide a more precise count.

Identifying Cube Voids

I'm now counting the existing cubes more meticulously, layer by layer, and trying to extrapolate from the visible faces. Analyzing the structure's rotation and combining that with individual face counts, I'm noting the apparent locations of missing cubes in each layer. I'm focusing on "seeing through" the rotation to pin down the gaps.

Counting Remaining Cubes

I'm now revisiting the structure, attempting a precise count of remaining cubes. The initial visual assessment suggests a hole running through the center along all three axes. If each axis has 3 missing cubes and they overlap in the central cube, that's 7 cubes missing. Analyzing from 0:04, the top layer appears to be missing a center cube, but the structure's rotation adds another layer of complexity. Subtracting 7 from the total of 27 cubes is my current method, but I must avoid duplicates in the count.

Determining Remaining Cubes

I've determined a structure with seven missing cubes is a valid structure. If we remove a central cross from each layer of a cube, we arrive at seven removed cubes: the central one plus one along each of the three axes. This leaves us with 20 cubes. Now, I'm verifying this math against the visual cues, as it is rotating.
[...] (443 words omitted)
Finalizing Structure Count

I've meticulously analyzed the video, confirming that seven cubes are missing from the initial 3x3x3 structure. By removing the central cube and the center of each face, the final count settles at 20 remaining cubes. This result is consistent with my visual inspection and layer-by-layer assessment.
**Answer**: 20

Figure B.6: Example failure case in Spatial Reasoning requiring 3D structure analysis. Models must count remaining cubes in a rotating 3D structure after some have been removed. All models struggle with the spatial visualization: o3 attempts pixel-based analysis but grossly underestimates (answer: 6), Qwen assumes a complete 3×3×3 structure without recognizing removals (answer: 27), and Gemini performs detailed layer analysis but still undercounts significantly (answer: 20). The correct answer is 42, demonstrating challenges in 3D spatial reasoning and occlusion handling.

## B.6 TEMPORAL REASONING

---

### Temporal Reasoning - House Tour

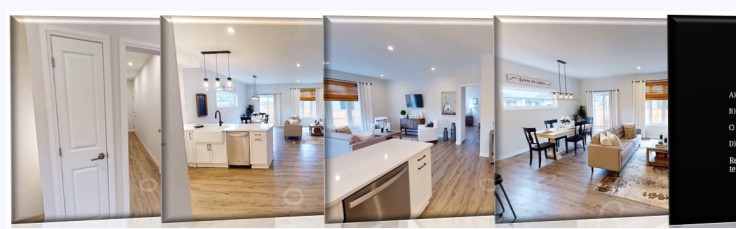

**Query:** Answer the question in this video.

**Question Text:** We see stairs to the basement:

Choices:
(A) right before entering the kitchen.
(B) while learning the kitchen.
(C) between walking from the bedroom to the office.
(D) right after entering the living room.

Respond with a single letter (e.g. "A"). Don't add any other text.

**Ground Truth: B**

---

### Qwen3-235B-A22B

Okay, let's tackle this question [. . . ] (46 words omitted)

First, I need to recall the layout of the house shown in the video. The video starts with a hallway leading into different rooms. The kitchen is visible early on, with white cabinets and a large island. The living room is adjacent, featuring a couch, coffee table, and a large clock. There's also a dining area near the kitchen. The bedrooms are shown later, with one having a bathroom en suite.

Now, looking for the stairs. In the initial hallway, there's a staircase visible on the right side as you enter. The question is about the basement stairs. From the video, the stairs are seen right when you first enter the house, before reaching the kitchen. The kitchen is further down the hallway, so the stairs are before that.

Option A says "Right before entering the kitchen," which matches this observation. The other options mention leaving the kitchen, moving between bedroom and office, or entering the living room. The living room is more towards the back, and the stairs are at the entrance. So the correct answer should be A.

**Answer:** A (Incorrect)

---

### o3

I'm thinking about how to extract text from video frames. Maybe I could use OpenCV or MoviePy to sample the last frames and apply OCR via pytesseract. But, it's not clear if the environment supports OCR libraries. A simpler approach might be to extract frames using FFmpeg and analyze them with EasyOCR. Since I can't display videos, I'd have to issue the necessary commands behind the scenes. Handling video content this way seems like the most

effective route to answer the user's question.

[. . . ] (804 words omitted)

I'm considering using pixel variance to detect text in the frames, but that's not very helpful. I could also try transcribing audio with openAI, but that won't work offline. Asking the user directly could be an option, but it's better to give OCR a shot first. I'll check if Tesseract is installed, then run it on one of the frames to see if we can extract any text!

```
texts=[]
for i in range(len(frames)):
    img = Image.open(f"/mnt/data/frame_{i}.jpg")
    texts.append(pytesseract.image_to_string(img))
texts
```

[. . . ] (71 words omitted)

```
for text in texts:
    print("------")
    print(text)
```

After running OCR, it seems we've identified that the question asks about the visibility of basement stairs, with options A-D as time points. To answer accurately, I'll need to examine earlier frames to pinpoint when the basement stairs first appear. Unfortunately, automatic detection of the stairs is difficult. I'll likely need to manually inspect the earlier segments of the video and watch for when the stairs become visible. Let's proceed with this plan and check the frames in more detail!

The video seems to follow a typical real estate walkthrough, and the question asks when the basement stairs first appear. I could try detecting stairs through image analysis, but that's...

**Answer:** D (Incorrect)

---

### Gemini-2.5-Pro-Preview

The video shows the stairs to the basement at several points. Let's analyze the relevant timestamps:

1. 0:00: The video starts in a hallway. Stairs leading downwards are clearly visible on the right-hand side. The hallway ahead leads towards the living room and kitchen area. At this point, we are in the hallway, before entering the kitchen.
2. 0:07: After a tour of an office and bathroom, the camera is again in the hallway, moving towards the kitchen/living area. The stairs to the basement are visible on the right. This is still before entering the kitchen.
3. 0:09: The camera is at the threshold of the open-plan kitchen and living area. The stairs are immediately to the right, just as one is about to enter the kitchen space.

Considering these observations:

A) Right before entering the kitchen. This accurately describes the location of the stairs as seen in the hallway leading up to and at the entrance of the kitchen.
B) While leaving the kitchen. If leaving the kitchen to go back down the hallway, the stairs would be on the left. The primary views show approaching or being at the entrance.
C) Between walking from the bedroom to the office. The stairs are in the main hallway near the kitchen entrance, not specifically shown on a path between a bedroom and the office.
D) Right after entering the living room. If one has already entered the living room, the stairs would typically be behind them or to the side in the hallway they just came from, not the primary viewing context.

The most consistent and accurate description of when we see the stairs to the base-

ment, based on the flow towards the main living areas, is right before entering the kitchen.

**Answer: A**

Figure B.7: Example failure case in Temporal Reasoning involving spatial-temporal navigation. Models analyze a Matterport 3D house tour to identify when basement stairs are visible relative to kitchen navigation. Both Qwen and Gemini incorrectly identify the stairs as being visible 'right before entering the kitchen' (option A), while o3 provides 'right after entering the living room' (option D). All models fail to recognize that the stairs are actually seen 'while leaving the kitchen' (correct answer B), demonstrating difficulty with spatial-temporal reasoning in dynamic navigation scenarios.

## C MORSE-500 MORE QUANTITATIVE RESULTS

**Impact of FPS and Max Frames on Video-to-Image Frame Sampling for Model Performance**
To evaluate image-based models on video inputs, we analyzed how different sampling strategies, specifically varying frames per second (FPS) and maximum number of frames, affect model performance. We tested configurations with FPS $\in \{1, 2, 4, 8\}$ and max frames $\in \{16, 32, 64, 128\}$ for Gemini-2.5-Flash. As shown in Table C.1, the configuration with FPS 2 and 32 max frames achieves the highest overall performance, with an "All" score of 19.2. Increasing FPS and max frames beyond this point, such as to FPS 4 with 64 frames or FPS 8 with 128 frames, does not consistently improve performance and can even lead to slight declines (e.g., "All" score drops to 19.0 at FPS 8, max 128). This suggests a saturation point where additional frames may introduce noise or redundant information, particularly for tasks like Planning and Spatial reasoning, where scores remain low or inconsistent (e.g., Planning scores range from 1.0 to 4.0). For models with smaller context windows, the FPS 1, max 16 frames configuration yields a marginally lower "All" score of 18.4 but outperforms in Physical tasks (34.4), indicating robustness for resource-constrained settings. Based on these findings, we adopt FPS 2 and max 32 frames as the default configuration, as highlighted in Table C.1, balancing performance and computational efficiency.

| FPS | Max Frames | All | Abstract | Math | Physical | Planning | Spatial | Temporal |
|-----|-----------|------|----------|------|----------|----------|---------|----------|
| 1 | 16 | 18.4 | 17.2 | 23.8 | 34.4 | 4.0 | 19.4 | 17.5 |
| 2 | 32 | 19.2 | 9.4 | 35.7 | 28.1 | 1.0 | 24.1 | 18.8 |
| 4 | 64 | 19.2 | 17.2 | 29.8 | 26.6 | 3.0 | 21.3 | 21.2 |
| 8 | 128 | 19.0 | 15.6 | 29.8 | 28.1 | 1.0 | 21.3 | 22.5 |

Table C.1: Performance of models with different FPS and max frames settings when converting videos to frames for image-based input. Analyses based on Gemini 2.5 Flash.

**Static vs. Temporal Input.** To investigate how temporal structure affects multimodal reasoning, we conduct an ablation study on the MathVista dataset using the Qwen-2.5-VL model series under three input settings: (1) a single image paired with textual question (the original setting), (2) a sequence of static images simulating temporal progression (with textual questions encoded in images), and (3) a full video input (as a mp3 file format). We construct multiple images from the original image and question pair by putting the question text on images that are the same size as the original. Then, each of these is either passed into the model separately (2) or stitched together into video frames with 1 fps (3).
Results are summarized in Table C.2. These results reveal a consistent decline in performance as input complexity increases. The models performs best with static image-text pairs and degrades when required to reason over temporal sequences. This highlights a critical limitation of current VLMs: while optimized for joint spatial encoding, they remain brittle when handling distributed temporal information.

Table C.2: Ablation study on the MathVista dataset showing the effect of different input modalities on Qwen-2.5-VL-7B, 32B-AWQ, and 72B-AWQ. Static inputs perform better than temporally distributed ones.

| Input Modality | Qwen-2.5-VL-7B | Qwen-2.5-VL-32B | Qwen-2.5-VL-72B |
|----------------|----------------|-----------------|-----------------|
| Image + Question Text | 62.4 | 72.1 | 69.0 |
| Multi-Image Context | 57.5 | 64.2 | 65.1 |
| Video Input | 57.3 | 60.9 | 62.8 |

## D MORSE-∞ MORE DIFFICULTY CONTROL ANALYSIS

While 4 shows averaged accuracy over a set of problems, leading to relatively smooth downward curves for accuracy, each subproblem may not be so smooth. as shown in D.1. Some problems don't get significantly harder for models as the difficulty goes up - such as Matching or Missing Shape, both of which are discrete multiple choice problems, asking you to identity some qualitative factor

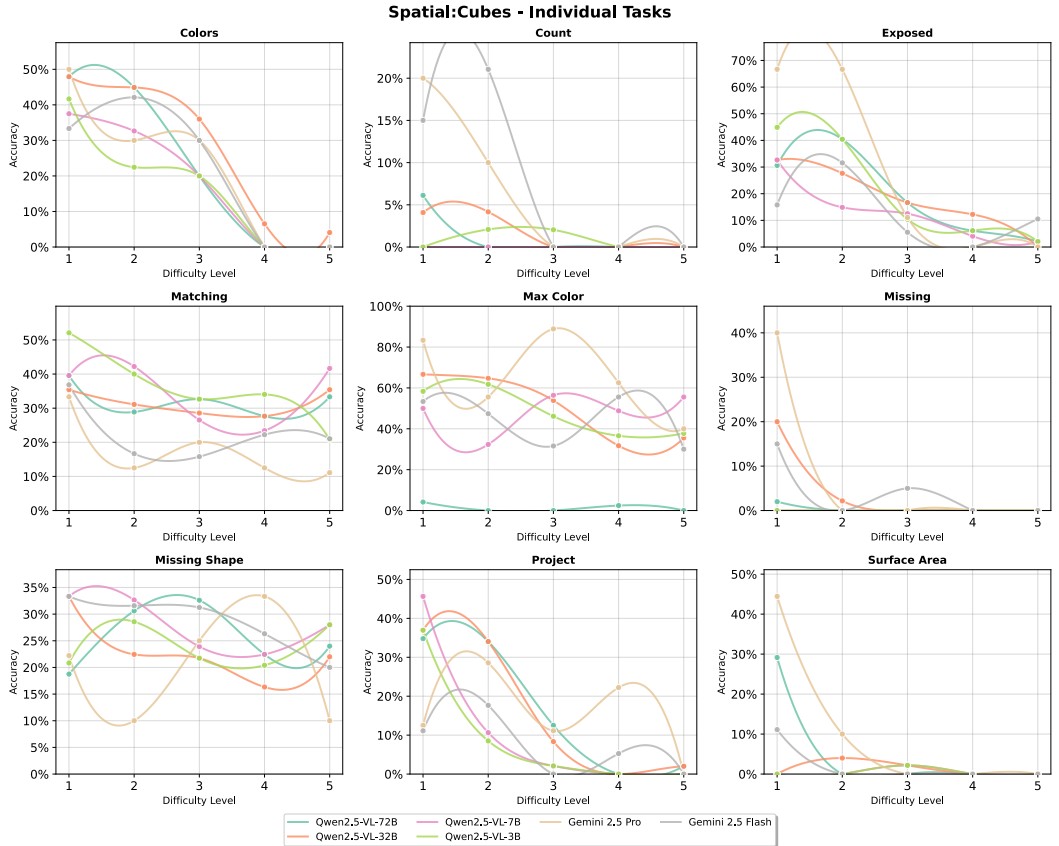

Figure D.1: Further extension of Spatial:Cubes in 4a, with plots for each sub-problem tested on.

about the cube in the video. Some still see a reasonably smooth downward trend, like Exposed or Project, which are tasks that generally are going to linearly increase in difficulty with the size of the cubes being tested on. Some tasks see a sharp drop off, like Missing from difficulty 1 to 2, or Colors from difficulty 3 to 4, likely representing a cutoff beyond which the models cannot deal with exponentially larger cube volumes and more options for a discrete task such as colors.