# OpenReview forum: "MORSE: A Suite of Programmatically Controllable Multimodal Reasoning Environments with Steerable Difficulty"
_ICLR.cc/2026/Conference — ICLR 2026 Conference Desk Rejected Submission_

### Official Review · Reviewer_EE2B · 2025-10-16

**Soundness:** 2
**Presentation:** 2
**Contribution:** 3
**Rating:** 2
**Confidence:** 4

**Summary:**

This paper present MORSE, the first integrated framework providing a unified programmatic ecosystem across training generation, question curation and autonomous verifications. The motivation for MORSE is to resolve the training and evaluation local optimal problem of multimodal large language model, where people use static imagery and in the meantime save human labor on annotating ground truth reasoning traces. MORSE integrates three components:

- MORSE-inifinite for scalable generation of question answer pairs with ground truth reasoning traces and steerable difficulty control.
- MORSE-500, which contains 500 challenging video question-answer instances including 6 reasoning categories. The authors also provide the evaluation results of multimodal language models on MORSE-500.
- MORSE-Agent which is an agentic framework to automatically writing new video generator. MORSE-Agent consists of Ideal proposal, initial implementation with python scripts and VLM as critic.

**Strengths:**

- The motivation for this paper is clear, the paper aims to solve the current limitations of current static development pipeline multi-modal langauge model training data and evaluation data.

- The originality of this paper is good, the paper introduces the first ecosystem for generating video, question and ground truth reasoning trace along each video. The ecosystem design is interesting.

- The authors evaluate MORSE-500 on 21 different models, that evaluation is sufficient and can indicates the consistent limitations of current MLLMs on challenging video tasks.

- The authors write their contributions very clear and the usage of bullet point and bold fonts, it is easy for readers to capture their contributions

**Weaknesses:**

Here I illustrate weakness using bullet points and followed by 1,2,3 for detailed illustrations.

- The originality of this paper is questionable. Although the authors claim they are the first ecosystem for generating video and question answer data for training and evaluation, many existing llm, mllm benchmark or training data also incorporate automatic data generation and question-answer using python scripts (like MORSE-infinite) and several filtering process using a powerful VLM (e.g. GPT or Gemini) (like MORSE-Agent), and in the end use human power for the last few rounds of evaluation and data filtering. So it is not a very new idea for automatic and programmatic mllm data generation, as it is already the preprocessing curation pipeline of many existing works.


- The data quality is the thing I really worry about. This is further divided into two questions:

       1. Instead of some traditional code generation or compensation tasks, this task incorporate automatic video generation using python scripts, which is a more difficult task which needs visual reasoning for verification. Normally for benchmark and even large scale training data there are human verification steps to ensure the overall quality. However in this work, the authors solely uses Gemini-2.5-Pro as critic for verification, although they refine a few steps, the abilities of Gemini-2.5-Pro is also limited and is not an almighty model. Although the authors claim their scalable or even infinite generation, I question the quality of the data can be used for training or evaluation.

       2. As shown in Table1, authors also test the Gemini-2.5-Pro's performance on MORSE-500, the accuracy of Gemini-2.5-Pro in most categories is lower than 30%, which is really low, so Gemini-2.5-Pro seems can not understand the question and answer the question. However authors use Gemini-2.5-Pro as their MORSE-Agent backbone for verification. I question the quality of the data to use a model as critic which can not even answer 30% of the questions without human-in-the-loop steps.

- The presentation of the paper is very brief and significantly lack convincing details. This can be further divided into four components:

       1. The paper significantly lacks detailed pipeline and examples and detailed component illustrations of MORSE-infinite, and MORSE-Agent. I understand that 9 page is not enough containing convincing details. But authors should contains links from the main content to the parts in appendix for detailed illustrations. Figure7 (the front of page 8) is not enough to brief the idea of MORSE-Agent, (e.g. what kind of code they are implementing, is there any self-debugging process for the MORSE-Agent, and how the interface designed for MORSE-Agent to generate videos, what kind of feedback do MORSE-Agent get from Gemini-2.5-Pro and how to build based on that, is it an end-to-end pipeline and when do the generation stop?) and can further make the quality of the generation pipeline questionable.

       2. The paper lacks illustrations about the task distribution and lacks detailed examples and illustrations. e.g. what is process understanding in temporal? Is the MORSE-infinite writing python scripts without handling simulation environment? e.g. In robot manipulation sub-task of planning category, how the data is curated?  If MORSE can handle the data generation within simulation environment, then what kind of simulation environment it is using?

       3. Authors claim the significance of ground truth reasoning trace of the agent, however in MORSE-500, the author doesn't use any reasoning traces to do any failure mode analysis of existing benchmarks. How the reasoning traces be used for analysis? Authors should provide examples and self-verifies the previous claims that they made.

       4. Experiment results on MORSE-500 lacks setting illustrations. e.g. which kind of prompt is used. How to sample frames from the video for evaluation

- Many models achieve very similar scores on MORSE-500. Especially for the planning category, it is hard to distinguish them. In Section 3.2, authors uses three paragraphs in enumerating the quantitative observations, but missing failure modes and also lacking insights on detailed analysis of why the model fail in reasoning. The author claims that they provide ground truth reasoning traces, but they didn't use it on their MORSE-500.

- The presentation of this paper is not very good. Figure 1 and Figure 8 is very vague to see clearly the text. Many picture in this paper is vacant and should be filled in more information and provide more visualizations on examples. Overall the paper seems a little bit rush and needs a lot of polishment.

**Questions:**

The motivation of this paper is good. But I feel like it needs a lot polishment for the authors to make it get accepted. In order to help authors with their rebuttal and modifications of the papers, I write the following bullet points for authors to consider:

For motivations:

- What specifically distinguishes your approach from existing MLLM benchmark/training data generation pipelines that also use Python scripts and VLM filtering?

For MORSE-infinite and MORSE-Agent:

- How do you justify using Gemini-2.5-Pro (with <30% accuracy on MORSE-500) as the sole critic backbone for MORSE-Agent? Since Gemini-2.5-Pro only achieves average less than 30% on MORSE-500. Apart from Gemini-2.5-Pro, do you also have other refinement steps to help the generation?
- Have you conducted any ablation studies to compare the data quality with and without human-in-the-loop verification?
- Is there any human verification in your MORSE-500 construction or MORSE-infinite pipeline design, if not, how can you guaranteed the generated samples qualified for training and evaluation?
- Have you done the generation evaluation of the MORSE-Agent, I feel like you should calculate the success rate of MORSE-Agent compared with existing code agent framework in order to distinguish them.
- Could you provide detailed algorithmic descriptions or pseudocode for MORSE-infinite and MORSE-Agent? Is MORSE-infinite uses sole python scripts and render videos through Manim/Maplotlib, does any other simulation environment involved? if not, how the robotic manipulation data is generated?
- What kind of feedback does MORSE-Agent receives from Gemini-2.5-Pro and how does it utilize it?
- What are the termination criteria for MORSE-Agent generation?

For MORSE-500:

- Why do many models achieve very similar scores, especially in the planning category? Since you can have steerable difficulty control, you should make the distribution uniform in order to distinguish different model's abilities
- Why do most models fail on these categories, can you provide insights on failure mode and future model improvements?
- What motivates in the category distribution, can you provide more insights on that?

---

> ### Comment · Reviewer_EE2B · 2025-11-27
>
> Hi Authors,
>
> If the rebuttal and revised version can clearly resolve the issues as mentioned above, I am willing to increase my score accordingly.
>
> Best,
>
> Reviewer EE2B

---

### Official Review · Reviewer_CGck · 2025-10-27

**Soundness:** 2
**Presentation:** 3
**Contribution:** 2
**Rating:** 4
**Confidence:** 5

**Summary:**

The paper introduces MORSE (Multimodal Reasoning Suite), a programmatically controllable collection of video reasoning environments with steerable difficulty and verifiable reasoning traces. MORSE comprises: (i) MORSE-\infty, a generator that produces unlimited, difficulty-controlled video tasks with ground-truth answers and stepwise traces; (ii) MORSE-500, a curated 500-video benchmark spanning six complementary reasoning categories; and (iii) MORSE-Agent, an agentic loop that proposes ideas, writes Python code to render videos, receives feedback from VLMs, and iteratively refines generation. On MORSE-500, the strongest model reaches 23.6% in accuracy, far below human performance (cf. 55.4%), with the largest gaps in abstract and planning reasoning.

**Strengths:**

- The paper proposes MORSE-\infty, a data simulator that produces unlimited, difficulty-steerable instances with reasoning traces and validated answers.
- The paper releases MORSE-500, a curated benchmark of 500 videos sampled from MORSE-\infty, spanning six complementary reasoning categories.
- The paper constructs MORSE-Agent, an agentic framework for automatically authoring new video generators.

**Weaknesses:**

- The presentation is good overall, but lacks clarity in some cases. For example, the text in Figure 1 is blurry, and the labels of Figure 5(b) overlap the radar plot. Moreover, in Table 1, it would be better to indicate the best-performing results to facilitate comparison.
- Lacking necessary details on the benchmark construction process. For example, it is unclear how the data simulator generates different types of tasks.
- Lacking novelty in evaluation tasks. Specifically, the considered evaluation tasks largely overlap with existing benchmarks, making it difficult to highlight novel contributions of this work. It would be better to address questions such as: What tasks can MORSE handle that other benchmarks cannot? And how can such a simulator effectively assist model development?

**Questions:**

Following the Weakness section, my questions are as follows:

Q1: Can you explain how the simulator generates the correct video for each specific task with a running example?

Q2: What tasks can MORSE handle that other benchmarks cannot? Since the evaluation data is generated in the video format, would it make a lot more sense to evaluate video understanding models?

Q2: For the MORSE-500 benchmark, how are the final answers evaluated? I am aware that you followed MathVista-how; but is this evaluation framework applicable to video-based tasks?

---

### Official Review · Reviewer_63Gq · 2025-10-28

**Soundness:** 2
**Presentation:** 2
**Contribution:** 2
**Rating:** 4
**Confidence:** 3

**Summary:**

MORSE is a suite of programmatically controllable video-reasoning environments that can generate unlimited problems with steerable difficulty and verifiable reasoning traces. It has three parts: (1) an infinite generator that emits videos with code-derived, step-by-step solutions; (2) MORSE-500, a benchmark spanning multiple reasoning categories; and (3) MORSE-Agent, which drafts new video generators by writing Python code and iterating with a VLM critic.

**Strengths:**

1.End-to-end controllable ecosystem: Unified generation → benchmarking → auto-authoring, with deterministic scripts and seeds for exact regeneration and trace-level supervision.

2.Steerable difficulty with verification: Explicit knobs for task complexity and validated reasoning chains; difficulty ramps reliably lower accuracy.

3.Breadth of reasoning: Six complementary categories—including temporal reasoning and planning—moving beyond static image QA to dynamic, multi-step video reasoning.

**Weaknesses:**

1. Synthetic bias & realism: The authors note that current generators emphasize synthetic environments; bringing in naturalistic scenarios while preserving verifiability is still open. Generalization to real‑world video remains uncertain.

2. Small public benchmark size: 500 items is modest for today’s models; while scripts/seeds help with regeneration, the fixed set could be studied/tuned against—risking overfitting unless the community routinely resamples.

3. Scope of “reasoning traces”: Traces are code‑derived (excellent for synthetic tasks), but the approach doesn’t directly provide verifiable chains for real‑world or unconstrained generative videos; extending verification beyond scripted worlds is non‑trivial.

**Questions:**

Refer to weakness

---

### Note · Program_Chairs · 2026-01-17
**Submission Desk Rejected by Program Chairs**

The following references in this submission do not refer to real documents and/or have major errors in bibliographic information:

 Pan Lu, Michel Galley, Hao Cheng, Kai-Wei Chang, and Jianfeng Gao. Learn from science textbooks: Retrieval-augmented science question answering. arXiv preprint arXiv:2207.05275, 2022. URL https://arxiv.org/abs/2207.05275.